# HiLoRA: high-frequency-augmented Low-Rank Adaptation

## Abstract

As large language models (LLMs) have demonstrated remarkable performance, parameter-efficient fine-tuning (PEFT) has emerged as an important paradigm. As a solution, low-rank adaptation (LoRA) freezes the pre-trained weights and introduces small learnable adapters instead of fine-tuning the full set of parameters. However, LoRA suffers from *catastrophic forgetting*, where pre-trained knowledge is overwhlemed and forgotten as new information is learned. One cause of this issue is *implicit regularization*, where deep learning models tend to favor more generalized solutions. This tendency leads to a significant increase in the largest singular values of the weights, which correspond to low-frequency components. To address this problem, we propose an advanced LoRA that balances the retention of pre-trained knowledge with the learning of new information. Since fine-tuning involves learning fine-grained details, which correspond to high-frequency information, we designed HiLoRA, a method that injects learnable high-frequency components into the pre-trained model. By leveraging the parameterized SVD and constraining singular values to appropriate levels, HiLoRA adapts to new tasks by focusing on the high-frequency domain with minimal change from the pre-trained weights. To evaluate the effectiveness of HiLoRA, we conduct extensive experiments on natural language understanding and question answering tasks. The results show that HiLoRA not only improves performance but also effectively retains pre-trained knowledge compared to baseline models.

## 1 Introduction

Pre-trained language models (PLMs) have achieved remarkable performance in various natural language processing tasks (Devlin et al., 2019; Liu et al., 2019; Lan et al., 2019; Radford et al., 2019; He et al., 2020; Touvron et al., 2023; Achiam et al., 2023; Anil et al., 2023). The common way to adapt pre-trained language models to downstream tasks is *fine-tuning*. However, fine-tuning all parameters of the model requires substantial resources. Especially, as the size of language models has grown to billions of parameters, storing copies of the large model for each downstream task results in significant memory consumption. To address this issue, recent studies suggest parameter-efficient fine-tuning (PEFT) methods (Hu et al., 2021; Zhang et al., 2023; Liu et al., 2024; Jiang et al., 2024; Meng et al., 2024; Wang et al., 2024), fine-tuning with only a small number of trainable parameters.

Low-Rank Adaptation (LoRA) (Hu et al., 2021), which updates parameters using low-rank matrices, has shown promising performance over other methods such as prompt tuning (Lester et al., 2021) or prefix tuning (Li & Liang, 2021). LoRA keeps the pre-trained weights frozen and updates only small number of parameters, which makes LoRA both storage- and compute-efficient. LoRA is designed based on the assumption that pre-trained language models are inherently low-dimensional and can learn efficiently even with random projections into smaller subspaces. The low-rank matrices serve as adapters, amplifying features that were learned but not emphasized during pre-training.

However, the LoRA-based fine-tuning methods have limitations. In general, deep learning-based models exhibit *implicit regularization*, a tendency for optimization algorithms and neural networks to favor more generalized solutions without overfitting (Arora et al., 2019; Cao et al., 2022; Zhao, 2022; Li et al., 2024). As a result, the larger singular values of the learnable weights tend to increase more significantly as training progresses (see Proposition 3.1). In Figure 1 (a), while the largest singular value increases during training, the test accuracy on pre-trained tasks decreases inversely.

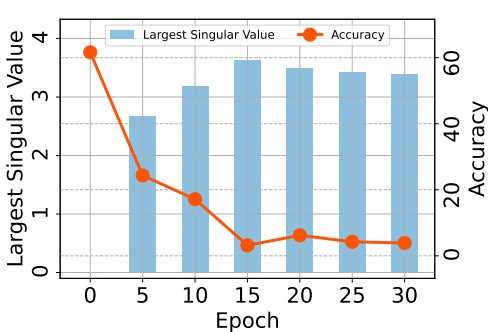

(a) Trade-off between the largest singular value and accuracy

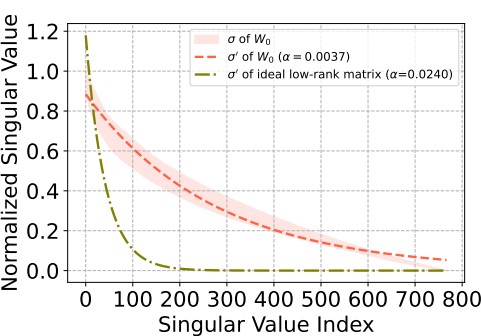

(b) Singular value decay

Figure 1: (a) The trade-off between the largest singular value and accuracy on the pre-trained task (BookCorpus) of LoRA fine-tuned on the STS-B dataset of the GLUE benchmark for RoBERTa$_{\text{base}}$, (b) the comparison of the fitted singular values $\sigma'$ from the exponential decay function with the normalized singular values for: i) the last output projection layer weight in the self-attention mechanism of DeBERTaV3$_{\text{base}}$, and ii) an ideal low-rank matrix with rank $r = 64$. Additionally, Figure 6 in Appendix C illustrates the low-rank approximation error rates for the two matrices.

This suggests that implicit regularization is also observed in LoRA, where the largest singular value increases during the fine-tuning process of the pre-trained model. Consequently, the low-frequency components corresponding to large singular values of the introduced modules exert a significant influence on the new task, overshadowing the pre-trained knowledge. The model gradually adapts to the new task as their dominance grows, leading to catastrophic forgetting where the pre-trained knowledge is overwhelmed and forgotten as the model learns new information.

To overcome this limitation, we propose a **Hi**gh-frequency augmented **Lo**w-**R**ank **A**daptation method, called **HiLoRA**, which effectively learns new knowledge while retaining the pre-trained knowledge. It is known that the low-frequency components are associated with large singular values and handle global information while high-frequency components correspond to smaller singular values and capture fine-grained details (Cooley et al., 1969; Deng & Cahill, 1993; Pan et al., 2022). The pre-trained model has already learned high-frequency components during its pre-training phase, and the high-frequency components contain valuable information, rather than simply representing noise. In Figure 1 (b), we compare the results of fitting an exponential decay function to the singular values in the pre-trained weights and those of an ideal low-rank matrix. The decay rate $\alpha$ of the weights of model is significantly smaller than that of the ideal low-rank matrix, indicating that the lower singular values retain relatively large magnitudes. Therefore, the high-frequency components in the pre-trained weights play a crucial role in retaining fine-grained information.

*Fine-tuning*, literally, is the process of injecting new knowledge on top of the pre-trained information, allowing the model to handle task-specific fine-grained details based on the major pre-trained information. Therefore, to efficiently fine-tune the pre-trained models, we propose to augment an appropriate level of high-frequency components into the pre-trained model through learnable modules. At this point, by limiting the singular values of the augmented components from becoming excessively large, the introduced modules can maintain its focus on the high-frequency domain. Through this process, the information is augmented in the high-frequency domain, allowing the model to effectively learn the new task while retaining its pre-trained knowledge with only minimal deviation. We conduct extensive experiments to evaluate the effectiveness of HiLoRA, demonstrating that it consistently outperforms LoRA and its variants across various tasks. Additionally, we assess catastrophic forgetting across multiple baseline models, showing that HiLoRA significantly mitigates the forgetting of pre-trained knowledge. Moreover, we achieved the outstanding results with introducing at most 12 new high-frequency components, which is negligible w.r.t. the original model size. Our key contributions can be summarized as follows:

- We propose a simple yet effective low-rank adaptation method, called HiLoRA, which balances the retention of pre-trained knowledge with the learning of new information and mitigates catastrophic forgetting problem in LoRA.
- As the fine-tuning process learns fine-grained information on top of pre-trained knowledge, we augment the model with high-frequency components using parameterized SVDs. This approach ensures that the introduced learnable module adapts to new tasks without over-whelming the pre-trained knowledge.
- We perform comprehensive experiments across various tasks, including both natural language understanding and question answering, demonstrating that HiLoRA outperforms baseline models and effectively mitigates catastrophic forgetting.

## 2    RELATED WORK & PRELIMINARIES

### 2.1    TRANSFORMERS

Transformers can be understood from two key submodules: multi-head attention (MHA) and feed-forward network (FFN). The MHA with $h$ parallel heads performs the attention function as follows:

$$\text{MHA}(X) = \text{Concat}(\text{head}_1, \ldots, \text{head}_h)W_o, \quad \text{head}_i = \text{Softmax}\left(\frac{XW_{q_i}(XW_{k_i})^\intercal}{\sqrt{d_k}}\right)XW_{v_i}, \quad (1)$$

where $W_o \in \mathbb{R}^{d \times d}$ is an output projection weight and $W_{q_i}, W_{k_i}, W_{v_i} \in \mathbb{R}^{d \times d_h}$ are query, key, and value projection weights for each head $i$. $d_h$ is typically set to $d/h$. FFN performs two linear transformations with a ReLU activation as follows:

$$\text{FFN}(X) = \text{ReLU}(XW_{f_1} + b_1)W_{f_2} + b_2, \quad (2)$$

where $W_{f_1} \in \mathbb{R}^{d \times d_m}$ and $W_{f_2} \in \mathbb{R}^{d_m \times d}$. These architectures enable a model to understand the language patterns and generate human-like texts in natural language processing.

### 2.2    LOW-RANK ADAPTATION

LoRA (Hu et al., 2021) suggests the low-rank update of the pre-trained weights by the product of two low-rank matrices. For $h = w_0 x$, the modified forward pass becomes:

$$h = W_0 x + \Delta W x = W_0 x + BA x, \quad (3)$$

where $W_0, \Delta W \in \mathbb{R}^{d_1 \times d_2}$, $A \in \mathbb{R}^{r \times d_2}$ and $B \in \mathbb{R}^{d_1 \times r}$ with $r \ll \{d_1, d_2\}$. $A$ is initialized with a random Gaussian initialization and B with zero, so $\Delta W = BA$ is zero at the beginning of training. After fine-tuning, the learnable adapter $\Delta W$ can be integrated into the pre-trained weight $W$ without modifying the original model architecture or adding any additional inference overhead.

**Directly modifying the components of $\Delta W$.**   Recent studies have used SVD to analyze the components of pre-trained weights. PiSSA (Meng et al., 2024) assumes that the principal components have important information and enables faster convergence by updating only the top $r$ principal components while keeping the residual parts fixed. However, PiSSA directly modifies the principal components of the original model weights $W_0$, altering the major information previously learned. This modification leads to catastrophic forgetting, where the pre-trained knowledge is forgotten during fine-tuning. Conversely, MiLoRA (Wang et al., 2024) proposes directly modifying the $r$ minor components, assuming that they are noisy and less important, in order to better preserve the pre-trained knowledge. However, existing models lack of consideration for changes in the frequency of introduced modules allows their influence to grow during fine-tuning, potentially overwhelming and forgetting the pre-trained knowledge.

**Adaptively adjusting the rank $r$.**   To adaptively adjust the rank $r$ in each layer, AdaLoRA (Zhang et al., 2023) parameterizes SVD and allocates the rank for each LoRA layer based on a sensitivity-driven importance score. SoRA (Ding et al., 2023) reduces the rank of each layer by introducing a sparsifying scheduler. These studies focus on pruning the number of ranks to meet a predefined budget using heuristic importance scores. However, these methods are designed for dynamically adjusting the rank of the weight, and does not consider its frequency structure.

**High rank update of $\Delta W$.** Recent studies focus on increasing the number of rank $r$ while avoiding any additional inference overhead. MoRA (Jiang et al., 2024) introduces a non-parametric operator that reduces the input dimension and increases the output dimension, allowing high-rank updates without significantly increasing the number of learnable parameters. The number of eigenvalues in the learnable adapters is larger, improving the capability of updates required to handle more complex tasks. ReLoRA (Lialin et al., 2023) achieves high-rank updates by leveraging the rank of sum property through multiple low-rank updates.

# 3 PROPOSED METHOD

## 3.1 MOTIVATIONS

In general, deep learning-based models demonstrate an *implicit bias*, a tendency for optimization algorithms and neural networks to favor simpler and more generalizable solutions without overfitting. The following theory explains the change of singular values of the learnable matrix $W$ in the absence of explicit regularization:

**Proposition 3.1** (Gradient descent induces large singular values via implicit regularization (Zhao, 2022)). *Under the assumptions specified in (Arora et al., 2019), the trajectory of the singular values $\sigma_n$ of the end-product matrix $W$ can be approximately characterized as:*

$$\dot{\sigma}_n = -vec(V_n U_n^\intercal)^\intercal P_{W,G} vec(\nabla_W \mathcal{L}(W)), \tag{4}$$

$$vec(\dot{W}) = -P_{W,G} vec(\nabla_W \mathcal{L}(W)), \tag{5}$$

*where $\dot{\sigma}$ is the derivative of $\sigma_n(t)$, $\{U_n, V_n\}$ are the left/right singular vectors of $W(t)$ corresponding to $\sigma_n(t)$, $N$ is the depth of network and $vec(\cdot)$ denotes vectorization. $P_{W,G} = \sum_{j=1}^{N} \left( (WW^\intercal)^{\frac{j-1}{N}} \otimes (W^\intercal W)^{\frac{N-j}{N}} \right) G_j$, where $G_j = diag(vec(S_j))$ is a positive semi-definite diagonal matrix for $j$-th layer, $[S_j] = \left( \nabla_{W_j} \mathcal{L}(W)^2 + s_j^2 \right)^{-1/2}$, $s_j^2 = var(\nabla_{W_j} \mathcal{L}(W))$ and $\nabla_{W_j} \mathcal{L}(W)$ is the loss gradient of $j$-th layer.*

Proposition 3.1 demonstrates that the large singular values of networks tend to become larger and the small singular values tend to become smaller. As revealed in (Zhao, 2022), the eigenvalue of $P_{W,G}$ is derived as $(1 + \eta^2)_{n,n'}^{-1/2}$ and dynamically adjusted based on the magnitude of the gradient and the weight scale. In the directions with the large singular values, the gradient magnitude is relatively large, resulting in a smaller $\eta^2$, which enhances the contribution of those directions. Conversely, in the directions of the small singular values, $\eta^2$ becomes larger, suppressing learning towards those directions. As depth increases, the weighted combination of the preconditioning matrices across layers accumulates, further emphasizing the directions of the large singular values and the gap among singular values becomes more distinct. However, as illustrated in Figure 1 (a), the tendency for singular values to increase as training progresses is directly related to the degradation of performance in pre-trained tasks. This phenomenon is called *catastrophic forgetting*, where the model forgets pre-trained knowledge as it learns new information, leading to the performance degradation on pre-trained task. Catastrophic forgetting hinders the continuous performance and consistency of LLMs, making it crucial to prevent this issue.

## 3.2 HIGH-FREQUENCY AUGMENTED LOW-RANK ADAPTATION

Our goal is to enable the model to effectively learn new tasks while retaining its pre-trained knowledge. We have identified that one of the critical causes of catastrophic forgetting in LoRA-based methods is the increase in the singular values of $\Delta W$ during fine-tuning for new tasks, which amplifies the influence of low-frequency components. Therefore, we aim to address this issue by effectively managing the frequency spectrum of the learned model.

In general, deep learning models including LoRA-based models, are biased towards the spectrum, called *spectral bias* (Cao et al., 2019; Rahaman et al., 2019), meaning that the original model $W_0$ tends to learn low-frequency information first and high-frequency information during the later stages of pre-training. Specifically, certain patterns with high-frequency information are learned based on the global patterns with low-frequency information during in later stages. Fine-tuning is the process

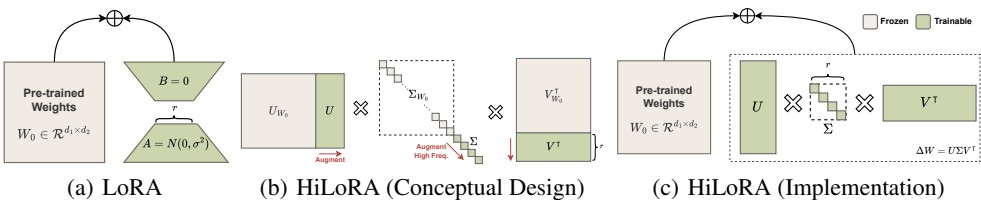

(a) LoRA      (b) HiLoRA (Conceptual Design)      (c) HiLoRA (Implementation)

Figure 2: The overall architectures of LoRA and HiLoRA in composing $W$. (a) Traditional LoRA, where the learnable adapter $\Delta W$ is treated as a residual adapter to the original weights $W_0$. (b) The conceptual illustration of HiLoRA, where $\Delta W$ represents new high-frequency components augmented into $W_0$. (c) The overall architecture of HiLoRA for implementation. HiLoRA does not directly decompose or reconstruct $W_0$ during fine-tuning.

of precisely adjusting the model to a new task based on the patterns learned during pre-training. Thus, the information during fine-tuning should be captured in the high-frequency domain. Specifically, to illustrate that the high-frequency components of the pre-trained model hold meaningful information rather than mere noise, we apply the Kolmogorov $n$-width (Pinkus, 2012) to the pre-trained weights. The Kolmogorov $n$-width measures how well complex data can be represented in an $n$-dimensional subspace. As shown in Figure 1 (b), the pre-trained weights have a much slower decay rate compared to an ideal low-rank matrix. This slower decay causes the singular values to decrease more gradually, making it difficult for the data to be fully represented in a small $n$-dimensional space. Consequently, the Kolmogorov $n$-width increases, indicating that the small singular values carry significant information.

Building upon this insight, we propose a high-frequency-augmented LoRA method. Figure 2 illustrates the difference in how traditional LoRA and our proposed HiLoRA handle $\Delta W$. While LoRA interprets $\Delta W$ as an adapter residual to the original pre-trained weights $W_0$, HiLoRA treats $\Delta W$ as an augmented high-frequency component to $W_0$. To define $\Delta W$ as a matrix of learnable components with appropriate frequency characteristics, we parameterize the introduced modules in the form of singular value decomposition as follows:

$$W = W_0 + \Delta W = W_0 + U\Sigma V^{\mathsf{T}}, \tag{6}$$

where $U \in \mathbb{R}^{d_1 \times r}$, $V^{\mathsf{T}} \in \mathbb{R}^{r \times d_2}$ are parameterized left/right singular vectors, respectively, and $\Sigma \in \mathbb{R}^r$ contains the parameterized singular values $\{\sigma_n\}_{1 \le n \le \min\{d_1, d_2\}}$. From the perspective of matrix operations, Equation 6 can be written as $W_0 + \Delta W = U_{W_0}\Sigma_{W_0}V_{W_0}^{\mathsf{T}} + U\Sigma V^{\mathsf{T}}$, where $U_{W_0}\Sigma_{W_0}V_{W_0}^{\mathsf{T}}$ represents the actual SVD of the pre-trained weights. This shows that new components $U\Sigma V^{\mathsf{T}}$ are augmented to the pre-trained weights $W_0$, as illustrated in Figure 2 (b). Note that SVD on $W_0$ is performed only once before fine-tuning to initialize $\bar{\sigma}$ whereas existing methods (Wang et al., 2024; Meng et al., 2024) extract singular vectors of $W_0$. The actual operation does not involve any explicit decomposition or reconstruction of $W_0$ during the fine-tuning process (see Appendix D). $U, V$ can be initialized with random $r$ singular vectors of $W_0$, or $U$ is initialized with zero and $V$ with a random Gaussian initialization. As mentioned earlier, we maintain the frequency components of the introduced modules at an appropriate level to prevent them from overwhelming the pre-trained knowledge. According to the definition of singular value decomposition, singular values must be non-negative, and we clamp them to the upper bound of augmented frequency $\bar{\sigma}$ to prevent the weights from becoming too large. This can be expressed by the following equation:

$$\sigma_n = \min(\max(\sigma_n, 0), \bar{\sigma}), \tag{7}$$

where $\bar{\sigma}$ can be set as a hyperparameter. The degenerate case of the proposed method occurs when all components hold the same information under the constraints on the parametrized singular values. This happens when the parameterized singular vectors align in the same direction, and all singular values converge to $\bar{\sigma}$, which significantly impacts the original model. Specifically, the maximum Frobenius norm of $\Delta W$, denoted as $\|\Delta W\|_F$, occurs when all singular values are equal to $\bar{\sigma}$. In this case, the Frobenius norm $\|\Delta W\|_F$ is given as $\|\Delta W\|_F = \sqrt{\sum_{n=1}^{r} \sigma_n^2} = \sqrt{r\bar{\sigma}^2} = \bar{\sigma}\sqrt{r}$. Thus, the maximum possible Frobenius norm of $\Delta W$ is $\bar{\sigma}\sqrt{r}$, representing the scenario where the matrix has been transformed to have all singular values equal to the upper bound $\bar{\sigma}$. This result implies that when the singular values are constrained by $\bar{\sigma}$, The Frobenius norm of $\Delta W$ may increase by up to

$\bar{\sigma}\sqrt{r}$ at most, which characterizes the degerate case in which the structure of $\Delta W$ has been fully altered by pushing all singular values to their upper bound. To prevent such degenerate cases, we ensure that the components of the learned $\Delta W$ do not capture the same information. We achieve this by applying orthogonal regularization to the singular vectors during training, forcing them to be orthogonal to each other and thus capturing distinct information. To enforce the orthogonality of $U$ and $V$, i.e., $U^{\mathsf{T}}U = VV^{\mathsf{T}} = I$, we apply the following regularization term:

$$R(U, V) = \|U^{\mathsf{T}}U - I\| + \|VV^{\mathsf{T}} - I\| \tag{8}$$

where $I \in \mathbb{R}^{r \times r}$ indicates an identity matrix. This regularization term is controlled by the orthogonal regularization coefficient $\gamma$. We verify the orthogonality of the parameterized singular vectors in Appendix F.2. We summarize the detailed algorithm in Algorithm 1.

---

**Algorithm 1** How to train HiLoRA

**Input:** Dataset $\mathcal{D}$; total iterations $T$; learning rate $\eta$, $\gamma$, $\bar{\sigma}$.
**for** $t = 1, \ldots, T$ **do**
$\quad \Sigma_k^{(t)} = \min(\max(\Sigma_k^{(t)}, 0), \bar{\sigma})$
$\quad W_k^{(t)} = W_0 + U_k^{(t)}\Sigma_k^{(t)}(V_k^{(t)})^{\mathsf{T}}$
$\quad$ Update $U_k^{(t+1)} = U_k^{(t)} - \eta\nabla_{U_k}(\mathcal{L}(U_k^{(t)}, \Sigma_k^{(t)}, V_k^{(t)}) + \gamma R(U_k^{(t)}, V_k^{(t)}))$
$\quad$ Update $V_k^{(t+1)} = V_k^{(t)} - \eta\nabla_{V_k}(\mathcal{L}(U_k^{(t)}, \Sigma_k^{(t)}, V_k^{(t)}) + \gamma R(U_k^{(t)}, V_k^{(t)}))$
$\quad$ Update $\Sigma_k^{(t+1)} = \Sigma_k^{(t)} - \eta\nabla_{\Sigma_k}\mathcal{L}(U_k^{(t)}, \Sigma_k^{(t)}, V_k^{(t)})$
**end**
**Output:** The fine-tuned parameters $\{U^{(T)}, \Sigma^{(T)}, V^{(T)}\}$, $W^{(T)} = W_0 + U^{(T)}\Sigma^{(T)}(V^{(T)})^{\mathsf{T}}$.

---

### 3.3 COMPARISON WITH LoRA-BASED METHODS

In this section, we highlight the distinctions between our approach and other LoRA-based methods.

**Directly modifying the components of $\Delta W$.** PiSSA (Meng et al., 2024) and MiLoRA (Wang et al., 2024) assume that the principal components contain the major information, while the minor components hold long-tail information or noise. They directly modify $r$ principal/minor components of $W_0$ to learn new information. However, after initializing $\Delta W$ with these $r$ components, no further constraints are applied during training. Due to implicit regularization, the principal components of $\Delta W$ grow larger, exerting greater influence over the original information and causing the model to forget pre-trained knowledge. In contrast, HiLoRA preserves the entire pre-trained model and injects new knowledge with generated high-frequency components through parameterized SVD, effectively retaining the pre-trained knowledge by regulating the influence of $\Delta W$ from becoming too large.

**Adaptively adjusting the rank $r$ & High rank update of $\Delta W$.** AdaLoRA (Zhang et al., 2023) and SoRA (Ding et al., 2023) are designed to dynamically prune the number of ranks in each layer using SVD for stable training. On the other hand, some approaches, such as MoRA (Jiang et al., 2024) and ReLoRA (Lialin et al., 2023), aim to increase the rank of $\Delta W$ to enhance model capacity and improve performance. While these methods primarily focus on the rank $r$ itself, our proposed method focuses on regulating the frequency of $\Delta W$ for a given predefined rank $r$, allowing to efficiently adapts to new tasks without altering the overall rank.

## 4 EXPERIMENTS

### 4.1 EXPERIMENTS ON NATURAL LANGUAGE UNDERSTANDING

#### 4.1.1 EXPERIMENTAL SETUP

We evaluate HiLoRA on the General Language Understanding Evaluation (GLUE) benchmark (Wang et al., 2018a), which includes 3 categories of natural language understanding tasks: i) single-sentence (CoLA and SST-2); ii) similarity and paraphrasing (MRPC, QQP, and STS-B); iii) natural language inference tasks (MNLI, QNLI, and RTE). For a fair comparsion, following

Table 1: Comparison of various methods with RoBERTa$_{base}$ on GLUE tasks with 4 different random seeds. Full results with standard deviations are provided in Appendix E.1.3.

| Method | MNLI | SST-2 | CoLA | QQP | QNLI | RTE | MRPC | STS-B | Avg. |
|---|---|---|---|---|---|---|---|---|---|
| LoRA | **87.95** | 94.81 | 63.95 | 90.95 | 92.75 | 79.96 | 89.22 | 90.84 | 86.30 |
| AdaLoRA | 87.23 | 94.95 | 61.35 | 89.74 | 92.52 | 81.59 | 89.22 | 90.60 | 85.90 |
| PiSSA | 87.94 | 94.47 | 64.17 | 90.99 | 92.45 | 76.99 | 89.89 | 90.87 | 85.97 |
| MiLoRA | **87.95** | 94.61 | 64.62 | **91.00** | 92.87 | 81.77 | 89.46 | 91.03 | 86.66 |
| HiLoRA | 87.94 | **95.10** | **64.66** | 90.73 | **93.12** | **82.85** | **90.20** | **91.16** | **86.97** |

Table 2: Comparison of various methods with DeBERTaV3$_{base}$ on SQuAD datasets

| | SQuADv1.1 | | | | | SQuADv2.0 | | | | |
|---|---|---|---|---|---|---|---|---|---|---|
| | 0.08% | 0.16% | 0.32% | 0.65% | **Avg.** | 0.08% | 0.16% | 0.32% | 0.65% | **Avg.** |
| Full FT* | 86.0 / 92.7 | | | | | 85.4 / 88.4 | | | | |
| HAdapter | 84.4/91.5 | 85.3/92.1 | 86.1/92.7 | 86.7/92.9 | 85.6/92.3 | 83.4/86.6 | 84.3/87.3 | 84.9/87.9 | 85.4/88.3 | 84.5/87.5 |
| PAdapter | 84.4/91.7 | 85.9/92.5 | 86.2/92.8 | 86.6/93.0 | 85.8/92.5 | 84.2/87.2 | 84.5/87.6 | 84.9/87.8 | 84.5/87.5 | 84.5/87.5 |
| LoRA | 86.4/92.8 | 86.6/92.9 | 86.7/93.1 | 86.7/93.1 | 86.6/93.0 | 84.7/87.5 | 83.6/86.7 | 84.5/87.4 | 85.0/88.0 | 84.4/87.4 |
| AdaLoRA | 87.2/93.4 | 87.5/93.6 | 87.5/93.7 | 87.6/**93.7** | 87.4/93.6 | **85.6/88.7** | **85.7/88.8** | 85.5/88.6 | **86.0/88.9** | 85.7/88.8 |
| HiLoRA | **87.9/93.8** | **88.0/93.9** | **88.0/94.0** | 87.7/93.7 | **87.9/93.8** | **85.6**/88.6 | **85.7**/88.6 | **85.7/88.7** | 85.8/88.8 | **85.7**/88.7 |

Hu et al. (2021), we adopt the pre-trained RoBERTa$_{base}$ as the backbone model. We use 1 GPU of NVIDIA RTX A6000 for experiments. We report Matthews correlation for CoLA, Spearman correlations for STS-B, and accuracy scores for the other tasks.

### 4.1.2 EXPERIMENTAL RESULT

Table 1 shows the experimental results of fine-tuning RoBERTa$_{base}$ on the GLUE task. MiLoRA, which freezes the low-frequency components while directly modifying the high-frequency components, showed the best performance among the other baselines. However, MiLoRA shows suboptimal performance due to the information loss caused by directly altering the high-frequency components in the pre-trained weights. However, HiLoRA shows the best average performance compared to other baselines, achieving the average accuracy of 86.97, Indicating that the new information of the fine-tuned dataset is effectively captured in the high-frequency components.

### 4.2 EXPERIMENTS ON QUESTION ANSWERING

### 4.2.1 EXPERIMENTAL SETUP

We evaluate HiLoRA on two question answering (QA) tasks: SQuAD v1.1 (Rajpurkar, 2016) and SQuADv2.0 (Rajpurkar et al., 2018). Following (Zhang et al., 2023), we fine-tune a pre-trained DeBERTaV3$_{base}$ (He et al., 2021) with HiLoRA and set the rank $r$ of LoRA as $\{2, 4, 6, 12\}$. These tasks are considered as a sequence labeling problem, where the goal is to predict the probability of each token being the start and end of the answer span. We measured the performance of model using the Exact Match (EM) and F1 metrics. We use 1 GPU of NVIDIA RTX 3090 24GB for experiments.

### 4.2.2 EXPERIMENTAL RESULT

Table 2 reports the experimental results on fine-tuning DeBERTa$_{base}$ on QA tasks. Both datasets showed significant improvements in average performance compared to full finetuning and LoRA. This suggests that the augmented high-frequency information played a crucial role. In particular, SQuADv1.1 exhibited notable improvements even with a small rank $r$. AdaLoRA and our model performed similarly on SQuADv2.0, indicating that each method plays a crucial role in different ways. AdaLoRA adapts by dynamically adjusting the rank $r$, while HiLoRA focuses on learning the high-frequency components with a fixed $r$. As a result, both methods optimize the model in different ways, leading to similar average outcomes, but with distinct advantages.

Table 3: The Frobenius norm of $U^\mathsf{T}WV$, where $U$ and $V$ are the left and right top $r$ singular vector directions of either: (1) $\Delta W_q$, (2) $W_q$, or (3) a random matrix. (4) The Frobenius norm of $U^\mathsf{T}\Delta WV$, where $U$ and $V$ are from $W_q$. (5) The Frobenius norm of $\Delta W$. (6,7) The introduced factors. The weights are taken from the last query layer of RoBERTa$_{\text{base}}$, fine-tuned on STS-B dataset with $r = 8$.

| Model | $\|U^\mathsf{T}WV\|_F$ | | | $\|U^\mathsf{T}_{W_q}\Delta WV_{W_q}\|_F$ | $\|\Delta W\|_F$ | $\text{Factor}_{W\to\Delta W}$ | $\text{Factor}_{\Delta W\to W}$ |
| | $\Delta W_q$ | $W_q$ | Random | | | | |
|---|---|---|---|---|---|---|---|
| LoRA | 0.48 | 11.22 | 0.32 | 0.16 | 3.81 | 7.94 | 23.82 |
| PiSSA | 0.38 | 11.22 | 0.35 | 0.11 | 2.49 | 6.54 | 22.60 |
| MiLoRA | 0.45 | 11.22 | 0.35 | 0.08 | 3.11 | 6.91 | 38.86 |
| HiLoRA | 0.36 | 11.22 | 0.38 | 0.03 | 0.94 | 2.60 | 31.18 |

## 5 ANALYSES ON HiLoRA

In this section, we aim to analyze the three characteristics of our model: i) the frequency analysis of $\Delta W$; ii) the relationship between $\Delta W$ and the pre-trained weights $W$; and iii) how HiLoRA effectively retains pre-trained knowledge while adapting to new tasks.

### 5.1 FREQUENCY ANALYSIS OF $\Delta W$

As shown in Proposition 3.1, the deep learning-based models exhibit implicit regularization, and the largest singular value increases as training progresses. To empirically validate that this tendency exists in LoRA-based methods, and that our proposed HiLoRA learns modules in the high-frequency domain, which leads to smaller singular values, Figure 3 illustrates the changes in the largest singular value of $\Delta W$ across various methods. While LoRA and its variants tend to increase the largest singular values as training progresses, our proposed HiLoRA maintains smaller singular values throughout training. This suggests that $\Delta W$ of HiLoRA primarily captures the information in the high-frequency domain.

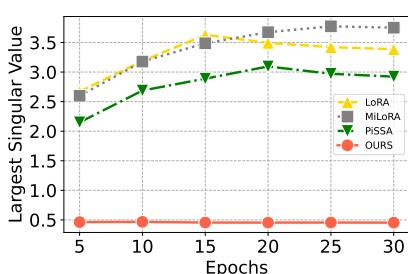

Figure 3: Largest singular value of $\Delta W$

### 5.2 HOW DOES THE ADAPTATION MATRIX $\Delta W$ COMPARED TO $W$?

We explore the relationship between $\Delta W$ and $W$ by measuring the correlation between $\Delta W$ and $W$ as well as the magnitude of $\Delta W$ in comparison to its corresponding directions in the pre-trained weight $W$. To do so, we introduce two key factors:

- $\text{Factor}_{W\to\Delta W}$ is a factor formulated as $\|\Delta W\|_F/\|U^\mathsf{T}_{\Delta W}WV_{\Delta W}\|_F$, which indicates the ratio of the norm of difference over the norm of projected $W$ on the $r$-dimensional subspace of $\Delta W$. This factor is also called *amplification factor* (Hu et al., 2021), measuring how the new information of $\Delta W$ is related to the existing information of $W$. A larger ratio refers that the task-specific information of $W$ has been amplified in $\Delta W$.

- $\text{Factor}_{\Delta W\to W}$ is a factor formulated as $\|\Delta W\|_F/\|U^\mathsf{T}_W\Delta WV_W\|_F$, which is the ratio of the norm of difference over the norm of projected $\Delta W$ on the $r$-dimensional subspace of $W$. It indicates the extent to which the change aligns with $W$. A larger ratio refers that $\Delta W$ has learned new information that is not present in $W$.

Following (Hu et al., 2021), we project $W$ onto the $r$-dimensional subspace of $\Delta W$ by computing $U^\mathsf{T}WV$, where $U$ and $V$ are the left and right singular vectors of $\Delta W$, $W$, and the random matrix. Additionally, we project $\Delta W$ onto the subspace of $W$ by computing $U^\mathsf{T}\Delta WV$. As shown in Table 3, HiLoRA and other methods exhibit similar Frobenius norms when $W$ is projected onto the subspace of $\Delta W$, $W$ and random matrix. However, compared to the baselines, the projection of $\Delta W$ onto the subspace of $W$ in HiLoRA shows the lowest correlation with a value of 0.02, which

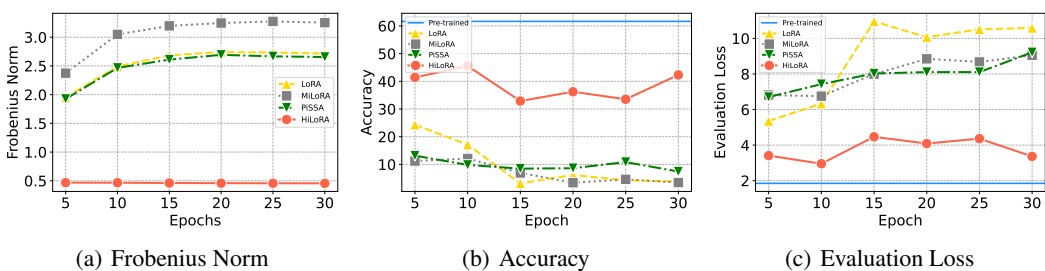

|(a) Frobenius Norm|(b) Accuracy|(c) Evaluation Loss|

Figure 4: Changes during fine-tuning RoBERTa$_{base}$ on the MRPC dataset of GLUE benchmark: (a) Frobenius norm of $\Delta W$ in the query layer, (b) accuracy on the pre-trained task (BookCorpus), and (c) evaluation loss on the pre-trained task.

is less than half of the smallest baseline. This suggests that HiLoRA processes the existing information in $W$ similarly to other methods, while being better at learning independent new information without relying on the existing information in $W$. Furthermore, considering the Frobenius norm of $\Delta W$, both LoRA and PiSSA exhibit a large $\texttt{Factor}_{W \to \Delta W}$ and a small $\texttt{Factor}_{\Delta W \to W}$, indicating that $\Delta W$ primarily amplifies information already present in $W$. MiLoRA also shows a large $\texttt{Factor}_{\Delta W \to W}$, but this results from the large magnitude of $\Delta W$, leading to significant changes from the pre-trained weights. In contrast, HiLoRA exhibits a relatively small $\texttt{Factor}_{W \to \Delta W}$ of 4.25 but a large $\texttt{Factor}_{\Delta W \to W}$ of 46.77. Given the small magnitude of $\Delta W$, this indicates that HiLoRA stands out for its ability to learn new information that is not already in $W$ with minimal deviation from the pre-trained weights.

### 5.3 How HiLoRA mitigates catastrophic forgetting

Unlike existing methods, we constrain the information on the new task to prevent it from overwhelming the pre-trained knowledge. To do, the injected frequency of $\Delta W$ has the upper bound of appropriate frequency value. In this section, we investigate how HiLoRA mitigates the catastrophic forgetting during fine-tuning. The magnitude of the change in weights is used to measure the change from pre-trained knowledge to new knowledge . Figure 4 (a) shows the evolution of the Frobenius norm with respect to the difference between original and learned weights during fine-tuning of RoBERTa on the STS-B dataset of GLUE task. LoRA and its variants show a rapid increase in change as the epochs increase. In contrast, HiLoRA maintains a constant level of change even with increasing epochs. Furthermore, Fig 4 (b) and (c) show the accuracy and evaluation loss on pre-trained knowledge from the BookCorpus  dataset, which is the source dataset for the pre-trained RoBERTa model. As the number of epochs increases, LoRA and its variants rapidly degrade the accuracy on the pre-trained knowledge, dropping from the original performance of 0.6 to below 0.1, and the loss function increases by about 5 times. As mentioned earlier, without restrictions on the frequency domain during training, the model undergoes significant changes, leading to catastrophic forgetting of the pre-trained knowledge. On the other hand, HiLoRA effectively mitigates this phenomenon, minimizing the performance degradation on the pre-trained task.

## 6 Additional Studies

As the sensitivity analysis, we examine the effects of $\bar{\sigma}$ and $\gamma$, with the results for $\gamma$ provided in Appendix F.1. In the ablation study, we investigate the impact of the augmented components.

### 6.1 Sensitivity study on the upper bound of augmented frequency $\bar{\sigma}$

We constraint the maximum value of the parameterized singular values with the hyperparameter $\bar{\sigma}$ to learn the augmented high-frequency components. To analyze the impact of $\bar{\sigma}$ on performance, we fine-tune the DeBERTaV3$_{base}$ model on the SQuADv2.0 dataset and report EM/F1 score according to $\bar{\sigma}$. In our experiments, $\bar{\sigma}$ holds the $q$-th quantile value of the singular values distribution of $W_0$, denoted as $\sigma^{(q)}$. As illustrated in Figure 5, the performance peaks when $\bar{\sigma} = \sigma^{(3)}$, indicating that $\bar{\sigma}$ at the appropriately small value level allows the model to optimally learn the augmented

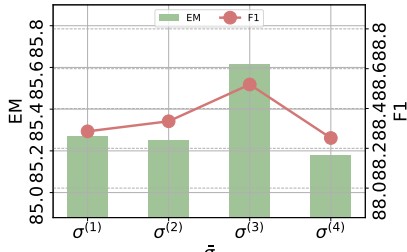

Figure 5: Sensitivity on $\bar{\sigma}$

| Model | MRPC | | SST-2 | |
|---|---|---|---|---|
| | $\text{Acc.}_{\text{fine-tune}}$ | $\text{Acc.}_{\text{pre-train}}$ | $\text{Acc.}_{\text{fine-tune}}$ | $\text{Acc.}_{\text{pre-train}}$ |
| Pre-trained | - | 61.64 | - | 61.64 |
| LoRA | 89.22 | 3.77 | 94.81 | 32.35 |
| $\text{LoRA}_{UV^\top}$ | 89.95 | 3.12 | 94.75 | 39.39 |
| $\text{LoRA}_{\text{SVD}}$ | 89.58 | 17.57 | 95.04 | 49.65 |
| HiLoRA | **90.20** | **32.00** | **95.10** | **51.29** |

Table 4: Ablation on the augmented components

high-frequency components while maintaining best accuracy. However, reducing or increasing $\bar{\sigma}$ too much leads to a degradation in both EM and F1 scores, suggesting that an inappropriate scale disrupts the capability of model to learn fine-grained details effectively.

## 6.2 ABLATION STUDY ON THE AUGMENTED COMPONENTS

To analyze the influence of the injected components in HiLoRA on the performance of both pre-trained and fine-tuned knowledge, we conduct an ablation study on the following variants: i) LoRA refers to the traditional LoRA method; ii) $\text{LoRA}_{UV^\top}$ applies orthogonal regularization to the singular vectors without considering the singular values; iii) $\text{LoRA}_{\text{SVD}}$ initializes the singular values as ones, allowing them to be learnable from $\text{LoRA}_{UV^\top}$; and iv) HiLoRA refers to the proposed method. We measure the accuracy on both the fine-tuned tasks, using the MRPC and SST-2 datasets from the GLUE benchmark, and the pre-trained task, using the BookCorpus dataset on $\text{RoBERTa}_{\text{base}}$. As reported in Table 4, LoRA significantly sacrifices pre-training performance to improve performance on fine-tuned tasks. For the MRPC dataset, accuracy on the pre-trained task drops from 61.64 to 3.77, while it achieves comparable accuracy on the fine-tuned task. $\text{LoRA}_{UV^\top}$ has limited expressiveness because its singular values are fixed at one. As a result, it may sacrifice either pre-trained or fine-tuned knowledge depending on the task. For the MRPC dataset, it outperforms LoRA but has lower accuracy on the pre-trained task, while for SST-2, it shows lower performance on the fine-tuned task but better accuracy on the pre-trained task compared to LoRA. $\text{LoRA}_{\text{SVD}}$ performs better due to its learnable singular values, enabling it to retain more pre-trained information than the original LoRA. Notably, HiLoRA constraints the singular values to learn in the high-frequency domain, ensuring both superior expressiveness and efficient retention of pre-trained information. For both datasets, HiLoRA achieves the best performance on both fine-tuned and pre-trained tasks.

## 7 CONCLUSION

We propose a simple yet effective low-rank adaptation method called HiLoRA, to address the problem of catastrophic forgetting in LoRA, where pre-trained knowledge is overwhelmed and forgotten as the model learns new information. Since fine-tuning incorporates fine-grained knowledge on top of the pre-trained information, we augment the pre-trained model with new high-frequency components, minimizing the impact on the pre-trained knowledge. HiLoRA achieves this by employing parameterized SVD and maintaining the augmented frequency components at appropriate levels. Our experimental results demonstrate that HiLoRA achieves promising performance on new tasks. Unlike traditional LoRA-based models, the learned models effectively capture high-frequency components and adapt to new information, rather than relying solely on pre-trained knowledge. With minimal changes, HiLoRA successfully integrates new information into the pre-trained weights, balancing between retaining pre-trained knowledge and adapting to new tasks.

**Limitations.** Despite the advantages of HiLoRA, there are a few limitations. First, while HiLoRA focuses on effectively capturing high-frequency components, in certain scenarios, it may under-represent some low-frequency information. Additionally, the optimal level of high-frequency components may vary across different datasets, requiring further tuning in some cases.

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

## A    REPRODUCIBILITY STATEMENT

In an effort to ensure reproducibility, we report the description of dataset in Appendices E.1.1 and E.2.1. Also we report the best hyperparameters of our experiments in Appendices E.1.2 and E.2.2. Our HiLoRA code to reproduce the experiment can be found at `https://bit.ly/4gGHlVs`.

## B    ETHICAL STATEMENT

We utilized publicly available datasets, including SQuAD and GLUE, which are commonly employed in academic research, and all sources have been appropriately cited. This research does not involve any personal or confidential information, thereby eliminating concerns related to privacy. Our proposed approach and the resulting insights contribute to the advancement of artificial intelligence while adhering to principles of ethical innovation and responsibility.

## C    EXPONENTIAL DECAY OF SINGULAR VALUES

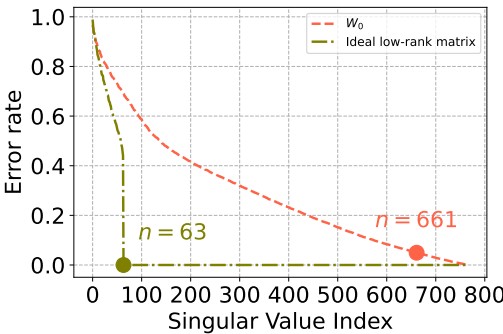

Figure 6: The error rate of the normalized singular values for: (i) the final output projection layer weights $W_0$ in the self-attention mechanism of DeBERTaV3$_{\text{base}}$, and (ii) an ideal low-rank matrix with rank $r = 64$. The marker indicates the $n$-value where the approximation error reaches 5%.

To find the best possible $n$-dimensional subspace $V_n$ such that the closest approximation $v \in V_n$ to $W$ minimizes the error $\|W - v\|_X$, the definition of Kolmogorov $n$-width is formulated as follows:

$$d_n(W, X) = \inf_{\substack{V_n \subset X \\ \dim V_n = n}} \inf_{v \in V_n} \|W - v\|_X, \tag{9}$$

where $V_n$ is $n$-dimensional subspace of $X$, $v$ is an element from the subspace $V_n$. 'inf' stands for infimum. When using the Frobenius norm (or spectral norm) with matrices, the Kolmogorov $n$-width is computed by the singular values of $W$ as follows:

$$d_n(W, X) = \sigma_{n+1}, \tag{10}$$

where $\sigma_{n+1}$ is the $(n + 1)$-th largest singular value of the matrix $W$. The Kolmogorov $n$-width measures how well a set $W$ can be approximated by an $n$-dimensional subspace. In other words, it represents the minimal maximum error when approximating with an $n$-dimensional subspace. Then we can determine the optimal dimensionality needed to achieve a desired approximation accuracy.

If the singular values decrease rapidly, $W$ can be well approximated even for small $n$, and the Kolmogorov $n$-width also decreases quickly. Therefore, the singular value decay rate $\alpha$, which plays a pivotal role in determining how effectively a matrix can be approximated, is commonly modeled by an exponential decay function as follows:

$$\sigma_n' = Ce^{-\alpha n}, \tag{11}$$

where $\sigma_n'$ represents the $n$-th modeled singular values, $C > 0$ is a constant, and $\alpha > 0$ is the decay rate. When the decay rate $\alpha$ is low, the singular values decrease gradually, resulting in large errors when approximating with the same $n$ dimensions. To minimize the approximation errors, a larger $n$ is required, indicating that significant information is contained in the lower singular values.

Figure 7: The overall architecture of HiLoRA for implementation. HiLoRA does not directly decompose or reconstruct $W_0$ during fine-tuning.

**Empirical analysis of the Kolmogorov $n$-width.** To empirically analyze the Kolmogorov $n$-width of the pre-trained language model, we present error rates based on low-rank approximation under the same conditions as shown in Figure 1 (b) of Introduction. The formulation of error rates $E_W(n)$ is as follows:

$$E_W(n) = \left( \frac{\|W - v\|_F}{\|W\|_F} \right) \times 100\%, \tag{12}$$

where $W$ is the original matrix and $v$ is the approximated matrix obtained by truncating the SVD to rank $n$. The error rates for the pre-trained model and the ideal low-rank matrix are presented in Figure 6, with markers indicating the $n$-value where the error rate reaches 5%. For the ideal low-rank matrix, the rank at which the error rate reaches 5% is 63. This suggests that the matrix has a low-dimensional structure, with the most important information concentrated in the top singular values. The lower singular values have little effect on the approximation and can be considered noise. In contrast, for the pre-trained model, the $n$-value required to reach 95% approximation is 661, which is significantly larger than ideal row rank matrix. This indicates that the data is complex and high-dimensional, and the lower singular values contain important information rather than merely noise.

# D AUGMENTATION OF THE NEW COMPONENTS

We augment the high-frequency components $\Delta W$ to the pre-trained weights $W_0$. From the perspective of matrix operations, the summation of two matrices can be regarded as augmenting new components as:

$$W = W_0 + \Delta W = U_{W_0} \Sigma_{W_0} V_{W_0}^\mathsf{T} + U\Sigma V^\mathsf{T} = [U_{W_0} \quad U] \begin{bmatrix} \mathrm{diag}(\Sigma_{W_0}) & 0 \\ 0 & \mathrm{diag}(\Sigma) \end{bmatrix} \begin{bmatrix} V_{W_0}^\mathsf{T} \\ V^\mathsf{T} \end{bmatrix}, \tag{13}$$

where $W_0 = U_{W_0} \Sigma_{W_0} V_{W_0}^\mathsf{T} \in \mathbb{R}^{d_1 \times d_2}$, where $U_{W_0} \in \mathbb{R}^{d_1 \times r}$, $\Sigma_{W_0} \in \mathbb{R}^{r \times r}$, and $V_{W_0}^\mathsf{T} \in \mathbb{R}^{r \times d_2}$, represent the singular vectors and singular values of the pre-trained weight matrix $W_0$. Note that, the singular value decomposition of $W_0$ is performed only once before fine-tuning to initialize $\bar{\sigma}$, and as illustrated in Figure 7, the actual implementation does not involve the explicit decomposition or reconstruction of $W_0$ during the fine-tuning process.

# E EXPERIMENTAL SETTINGS

## E.1 NATURAL LANGUAGE UNDERSTANDING

### E.1.1 DATASET DESCRIPTION

We describe the benchmark datasets of GLUE (Wang et al., 2018a) below.

- **CoLA.** The Corpus of Linguistic Acceptability (Warstadt et al., 2019) provides a dataset of English sentences, where each sentence is judged for grammatical acceptability based on

data from books and journal articles. The objective is a binary classification to determine whether a sentence is grammatically correct or incorrect. The dataset consists of 8.5k samples for training, 1k samples for validation, and 1k samples for test.

- **SST-2.** The Stanford Sentiment Treebank (Socher et al., 2013) includes sentences from movie reviews, along with human-provided sentiment annotations. The goal is to classify the sentiment of each sentence as either positive or negative. The dataset consists of 67k samples for training, 872 samples for validation, and 1.8k samples for test.

- **MRPC.** The Microsoft Research Paraphrase Corpus (Dolan & Brockett, 2005) contains pairs of sentences automatically extracted from online news sources. Human annotators label each pair, and the task is to identify whether the two sentences in a pair convey the same meaning. The dataset consists of 3.7k samples for training, 408 samples for validation, and 1.7k samples for test.

- **QQP.** The Quora Question Pairs dataset (Chen et al., 2018) consists of question pairs taken from Quora, a community-driven question-and-answer platform. The task is to determine if two given questions are semantically identical. The dataset consists of 364k samples for training, 40k samples for validation, and 391k samples for test.

- **MNLI.** The Multi-Genre Natural Language Inference Corpus (Williams et al., 2017) includes sentence pairs with textual entailment annotations collected through crowdsourcing. Given a premise and a hypothesis, the task is to predict whether the premise entails the hypothesis, contradicts it, or is neutral. The dataset includes both in-domain and cross-domain evaluations using a hidden test set. The dataset consists of 393k samples for training, 20k samples for validation, and 20k samples for test.

- **QNLI.** The Question-Answering Natural Language Inference dataset (Wang et al., 2018b) consists of question-paragraph pairs from which an answer must be found. The task involves determining whether a specific sentence from the paragraph answers the corresponding question. The dataset consists of 108k samples for training, 5.7k samples for validation, and 5.7k samples for test.

- **RTE.** The Recognizing Textual Entailment dataset (Bentivogli et al., 2009) comes from a series of annual challenges focusing on textual entailment. The task is to classify sentence pairs as either entailment or non-entailment. The dataset consists of 2.5k samples for training, 276 samples for validation, and 3k samples for test.

- **STS-B.** The Semantic Textual Similarity Benchmark (Cer et al., 2017) features sentence pairs drawn from various sources, including news headlines and image captions, with human-assigned similarity scores. The task is a regression problem where the model must predict a similarity score ranging from 0 to 5. The dataset consists of 7k samples for training, 1.5k samples for validation, and 1.4k samples for test.

### E.1.2 HYPERPARAMETERS

To tune HiLoRA, We search for the learning rate from $\{4 \times 10^{-4}, 5 \times 10^{-4}\}$, $\bar{\sigma}$ from $\{\sigma^{(2)}, \sigma^{(3)}, \sigma^{(4)}\}$ and $\gamma$ from $\{1 \times 10^{-1}, 7 \times 10^{-2}, 5 \times 10^{-2}, 3 \times 10^{-2}, 1 \times 10^{-2}, 1 \times 10^{-3}\}$. The learnable singular vectors $U/V$ can be initialized as i) random $r$ singular vectors of $W_0$, ii) $U$ with zero, $V$ with random Gaussian initialization. We report the best hyperparameters of HiLoRA in Table 5 below.

### E.1.3 EXPERIMENTAL RESULT WITH STANDARD DEVIATIONS

We report the experimental results on GLUE tasks with standard deviation in Table 6.

### E.2 QUESTION ANSWERING

### E.2.1 DATASET DESCRIPTION

We describe the benchmark dataset of SQuAD (Rajpurkar, 2016; Rajpurkar et al., 2018). The Stanford Question Answering Dataset (SQuAD) is a benchmark for reading comprehension, featuring questions based on Wikipedia articles. Each question is answered with a specific text segment (or span) from the corresponding passage, though some questions may have no answer at all.

Table 5: Best hyperparameters for HiLoRA in natural language understanding

| Dataset | Learning rate | Batch size | #Epochs | Metric | $\bar{\sigma}$ | $\gamma$ | How to initialize $U, V$ |
|---------|---------------|------------|---------|--------|----------------|----------|--------------------------|
| CoLA | $4 \times 10^{-4}$ | 32 | 25 | Matthews correlation | $\sigma^{(2)}$ | $3 \times 10^{-2}$ | random $r$ singular vectors |
| MNLI | $5 \times 10^{-4}$ | 32 | 7 | Accuracy | $\sigma^{(2)}$ | $1 \times 10^{-1}$ | 0, random Gaussian |
| MRPC | $4 \times 10^{-4}$ | 16 | 30 | Accuracy | $\sigma^{(2)}$ | $7 \times 10^{-2}$ | random $r$ singular vectors |
| QNLI | $4 \times 10^{-4}$ | 32 | 5 | Accuracy | $\sigma^{(2)}$ | $1 \times 10^{-2}$ | random $r$ singular vectors |
| QQP | $5 \times 10^{-4}$ | 32 | 5 | Accuracy | $\sigma^{(2)}$ | $1 \times 10^{-3}$ | 0, random Gaussian |
| RTE | $5 \times 10^{-4}$ | 32 | 50 | Accuracy | $\sigma^{(2)}$ | $5 \times 10^{-2}$ | 0, random Gaussian |
| SST-2 | $5 \times 10^{-4}$ | 32 | 24 | Accuracy | $\sigma^{(3)}$ | $1 \times 10^{-2}$ | 0, random Gaussian |
| STS-B | $4 \times 10^{-4}$ | 32 | 25 | Pearson correlation | $\sigma^{(3)}$ | $1 \times 10^{-1}$ | 0, random Gaussian |

Table 6: Comparison of various methods on GLUE tasks with 4 different random seeds.

| Method | MNLI | SST-2 | CoLA | QQP | QNLI | RTE | MRPC | STS-B | Avg |
|--------|------|-------|------|-----|------|-----|------|-------|-----|
| LoRA | **87.95**±0.13 | 94.81±0.10 | 63.95±1.08 | 90.95±0.04 | 92.75±0.10 | 66.88±14.20 | 89.22±0.30 | 90.84±0.09 | 86.30 |
| AdaLoRA | 87.94±0.01 | 94.47±0.20 | 64.17±1.19 | 90.99±0.05 | 92.45±0.18 | 73.01±10.70 | 89.89±1.23 | 90.87±0.17 | 85.90 |
| PiSSA | 87.23±0.05 | 94.95±0.34 | 61.35±0.87 | 89.74±0.09 | 92.52±0.08 | 81.59±1.11 | 89.22±0.39 | 90.60±0.10 | 85.97 |
| MiLoRA | **87.95**±0.16 | 94.61±0.29 | 64.62±0.99 | **91.00**±0.05 | 92.87±0.24 | 81.77±1.29 | 89.46±0.30 | 91.03±0.13 | 86.66 |
| HiLoRA | 87.94±0.11 | **95.10**±0.15 | **64.66**±0.65 | 90.76±0.07 | **93.12**±0.11 | **82.85**±0.79 | **90.20**±0.52 | **91.16**±0.21 | **86.97** |

- **SQuADv1.1.** Over 100,000 question-answer pairs derived from more than 500 articles. The dataset consists of 87,599 samples for training and 10,570 for validation.

- **SQuADv2.0.** Combines the 100,000 questions in SQuADv1.1 with over 50,000 unanswerable questions to closely resemble answerable ones. To perform well on SQuADv2.0, systems must not only provide correct answers when available but also recognize when a question cannot be answered based on the given passage and abstain from responding. The dataset consists of 130,319 samples for training and 11,873 for validation.

### E.2.2 HYPERPARAMETERS

To tune HiLoRA, We search for the learning rate from $\{1 \times 10^{-3}, 5 \times 10^{-3}\}$, $\bar{\sigma}$ from $\{\sigma^{(2)}, \sigma^{(3)}, \sigma^{(4)}\}$ and $\gamma$ from $\{7 \times 10^{-1}, 5 \times 10^{-1}, 1 \times 10^{-1}, 7 \times 10^{-2}, 5 \times 10^{-2}, 1 \times 10^{-2}\}$. The learnable singular vectors $U/V$ can be initialized as i) random $r$ singular vectors of $W$, ii) $U$ with zero, $V$ with random Gaussian initialization. We report the best hyperparameters of HiLoRA in Table 7 below.

Table 7: Best hyperparameters for HiLoRA in question answering

| Dataset | Learning rate | Batch size | #Epochs | Metric | $\bar{\sigma}$ | $\gamma$ | How to initialize $U, V$ |
|---------|---------------|------------|---------|--------|----------------|----------|--------------------------|
| | $1 \times 10^{-3}$ | | 10 | | $\sigma^{(2)}$ | $1 \times 10^{-2}$ | 0, random Gaussian |
| SQuADv1.1 | $1 \times 10^{-3}$ | 16 | 10 | EM/F1 | $\sigma^{(2)}$ | $1 \times 10^{-2}$ | 0, random Gaussian |
| | $1 \times 10^{-3}$ | | 10 | | $\sigma^{(2)}$ | $1 \times 10^{-1}$ | 0, random Gaussian |
| | $1 \times 10^{-3}$ | | 10 | | $\sigma^{(2)}$ | $5 \times 10^{-1}$ | 0, random Gaussian |
| | $1 \times 10^{-3}$ | | 12 | | $\sigma^{(3)}$ | $5 \times 10^{-1}$ | 0, random Gaussian |
| SQuADv2.0 | $5 \times 10^{-3}$ | 16 | 12 | EM/F1 | $\sigma^{(3)}$ | $5 \times 10^{-1}$ | 0, random Gaussian |
| | $1 \times 10^{-3}$ | | 12 | | $\sigma^{(2)}$ | $1 \times 10^{-1}$ | random $r$ singular vectors |
| | $1 \times 10^{-3}$ | | 12 | | $\sigma^{(3)}$ | $7 \times 10^{-2}$ | 0, random Gaussian |

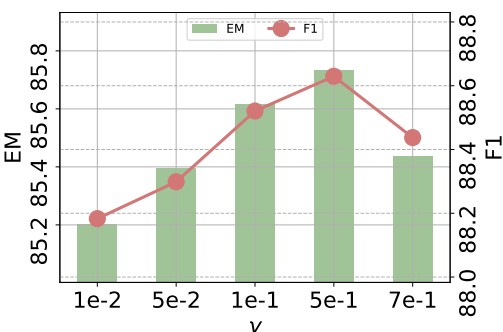

Figure 8: Sensitivity study on $\gamma$

# F ADDITIONAL STUDIES

## F.1 SENSITIVITY STUDY ON THE ORTHOGONAL REGULARIZATION COEFFICIENT $\gamma$

The orthogonal regularization applied to $U$ and $V$ is used to learn the singular values that consists the augmented high-frequency components. We further conduct sensitivity study on the effect of the orthogonal regularization coefficient $\gamma$. We fine-tuned the DeBERTaV3$_{base}$ model on the SQuADv2.0 dataset. As shown in Figure 8, appropriate regularization induces the orthogonalization of singular values, leading to improved convergence during fine-tuning and enhanced performance. However, excessive regularization results in performance degradation, indicating the need for an optimal balance that maximizes the benefits of regularization without hindering the ability of model to learn task-specific patterns.

## F.2 ORTHOGONAL REGULARIZATION ON PARAMETERIZED SINGULAR VECTORS

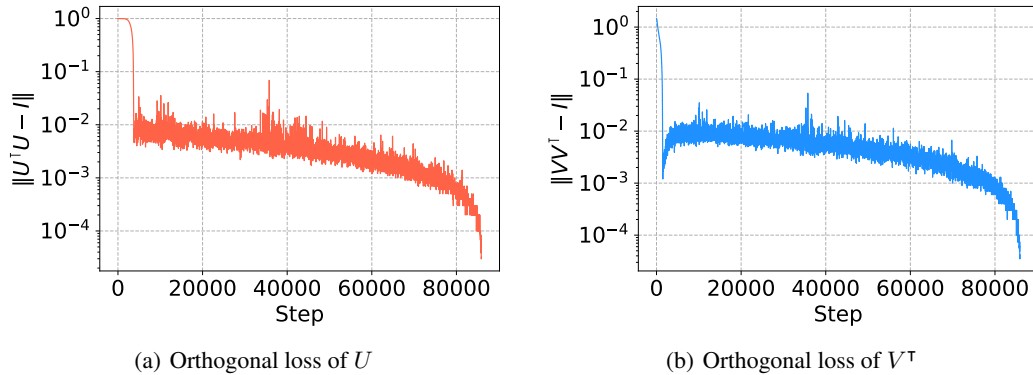

(a) Orthogonal loss of $U$          (b) Orthogonal loss of $V^{\mathsf{T}}$

Figure 9: The orthogonal loss curves of parameterized singular vectors $U$ and $V$ when fine-tuning RoBERTa$_{base}$ on STS-B dataset

Figure 9 shows the orthogonal loss curve of parameter singular vectors $U$ and $V$ of RoBERTa$_{base}$ fine-tuned on STS-B dataset. The singular vectors are orthogonally optimized as indicated by the consistent reduction in orthogonal loss throughout the fine-tuning process.

# G COMPARISON OF COMPUTATIONAL COMPLEXITY

Table 8 summarizes the empirical training time (min per epoch) and peak GPU usage (GB) of RoBERTa$_{base}$ fine-tuned on GLUE tasks. The GPU usage showed a very slight increase compared to the original LoRA, and the additional runtime occurs in HiLoRA and AdaLoRA. This increase

arises from the orthogonal regularization of singular vectors generated by parameterized SVD. However, fine-tuning typically requires fewer epochs, and considering the improved performance and the ability to retain pre-trained knowledge compared to the baseline model, this increase is negligible.

Table 8: Comparison of "training time (min per epoch)/peak GPU usage (GB)"

| Method | MNLI | SST-2 | CoLA | QQP | QNLI | RTE | MRPC | STS-B |
|--------|------|-------|------|-----|------|-----|------|-------|
| LoRA | 105.9/24.9 | 18.2/24.9 | 2.3/24.9 | 98.1/24.9 | 28.3/24.9 | 0.7/24.9 | 1.0/12.5 | 1.6/24.9 |
| PiSSA | 106.2/24.9 | 18.1/24.9 | 2.3/24.9 | 98.1/24.9 | 28.2/24.9 | 0.7/24.9 | 1.0/12.5 | 1.5/24.9 |
| AdaLoRA | 123.4/25.6 | 21.1/25.6 | 2.7/25.6 | 114.4/25.6 | 33.1/25.6 | 0.8/25.6 | 1.3/13.1 | 1.8/25.6 |
| MiLoRA | 106.0/24.9 | 18.1/24.9 | 2.3/24.9 | 98.1/24.9 | 28.2/24.9 | 0.7/24.9 | 1.0/12.5 | 1.5/24.9 |
| HiLoRA | 128.9/25.2 | 22.1/25.2 | 2.8/25.2 | 119.4/25.2 | 34.3/25.2 | 0.8/25.2 | 1.3/12.8 | 1.9/25.2 |

## H  CATASTROPHIC FORGETTING IN ADALORA

We measure the catastrophic forgetting phenomenon in AdaLoRA using the MRPC and STS-B datasets in the GLUE task. Specifically, we measure the largest singular value and Frobenius norm of the difference between the pre-trained model and the fine-tuned model. Also, we evaluate the accuracy and the evaluation loss on the pre-trained task, denoted as 'Acc.$_{\text{pre-train}}$' and 'Eval. loss$_{\text{pre-train}}$'. For each dataset, the metrics are measured every 5 epochs during the fine-tuning of AdaLoRA, and the metrics at the point where AdaLoRA and HiLoRA achieved their best accuracy are also reported, respectively, denoted as 'Best' and 'Best$_{\text{HiLoRA}}$'.

Table 9: Catastrophic forgetting in AdaLoRA fine-tuned on MRPC dataset

| Epoch | 5 | 10 | 15 | 20 | 25 | 30 | Best | Best$_{\text{HiLoRA}}$ |
|-------|---|----|----|----|----|----|------|------------------------|
| Largest singular value | 2.0177 | 2.0069 | 2.0016 | 2.0009 | 2.0004 | 2.0001 | 2.0177 | 0.9358 |
| Frobenius norm | 2.0201 | 2.0050 | 2.0014 | 2.0011 | 2.0005 | 2.0001 | 2.0201 | 0.9284 |
| Acc.$_{\text{pre-train}}$ | 25.578 | 14.879 | 8.962 | 16.794 | 10.591 | 11.343 | 25.58 | 32.00 |
| Eval. loss$_{\text{pre-train}}$ | 5.231 | 6.696 | 7.617 | 6.178 | 7.103 | 7.007 | 5.2308 | 4.3496 |

Table 10: Catastrophic forgetting in AdaLoRA fine-tuned on STS-B dataset

| Epoch | 5 | 10 | 15 | 20 | 25 | Best | Best$_{\text{HiLoRA}}$ |
|-------|---|----|----|----|----|------|------------------------|
| Largest singular value | 2.0064 | 2.0034 | 2.0056 | 2.0021 | 2.0001 | 2.0001 | 0.9351 |
| Frobenius norm | 2.0087 | 2.0030 | 2.0065 | 2.0013 | 2.0001 | 2.0001 | 0.9312 |
| Acc.$_{\text{pre-train}}$ | 33.526 | 34.118 | 29.61 | 28.19 | 28.485 | 28.49 | 43.72 |
| Eval. loss$_{\text{pre-train}}$ | 3.984 | 3.951 | 4.426 | 4.566 | 4.525 | 4.5248 | 3.1785 |

According to Tables 9 and 10, AdaLoRA also experiences catastrophic forgetting as its fine-tuning progresses. The Frobenius norm increases from 0 to 2 in the early fine-tuning phase, with the performance on the pre-trained task decreases. Even at its peak performance during fine-tuning, the model still exhibits low performance on the pre-trained task. This can be attributed to the lack of consideration for frequency components of adapters, leading to a tendency for learning the low-frequency components while forgetting the pre-trained information. In contrast, the proposed model regularizes the frequency components in the adapter, injecting the new knowledge into high-frequency components during fine-tuning. As a result, the proposed model retains pre-trained information more effectively.

## I  LARGE-SCALE EXPERIMENTS ON COMMONSENSE REASONING

We conduct the experiments for the commonsense reasoning task on LLaMA-7B. Following (Hu et al., 2023), we amalgamate the training datasets from 8 sub-tasks to create the final training dataset, and conduct evaluations on the individual testing dataset for each task.

Table 11: Accuracy comparison on eight commonsense reasoning datasets

| LLaMA-7B | BoolQ | PIQA | SIQA | HellaSwag | WinoGrande | ARC-e | ARC-c | QBQA | Avg. |
|---|---|---|---|---|---|---|---|---|---|
| LoRA | 68.9 | 80.7 | 77.4 | 78.1 | 78.8 | 77.8 | 61.3 | 74.8 | 74.7 |
| HiLoRA | 62.2 | 82.7 | 78.3 | 81.0 | 80.9 | 83.3 | 66.8 | 78.6 | **76.7** |

As reported in Table 11, HiLoRA demonstrates improved performance over LoRA on average in large-scale models, highlighting its stability and effectiveness while maintaining strong results across diverse downstream tasks.

## J EFFECT OF RANK $r$ ON CATASTROPHIC FORGETTING

To verify whether HiLoRA maintains its performance and continues to mitigate forgetting as the rank increases, we conduct sensitivity study on the rank $r$ on MRPC and STS-B dataset. Specifically, we measured the largest singular value and Frobenius norm of the difference between the pre-trained model and the fine-tuned model. Also, we evaluated the accuracy and the evaluation loss on the pre-trained task, and accuracy on the fine-tuned task.

Table 12: The effect of rank $r$ on catastrophic forgetting

| $r$ | Metric | 8 | 16 | 64 |
|---|---|---|---|---|
| MRPC | Largest singular value | 0.93 | 0.69 | 0.36 |
| | Frobenius norm | 0.94 | 0.70 | 0.36 |
| | Eval. loss$_{\text{pre-train}}$ | 4.35 | 4.18 | 3.20 |
| | Acc.$_{\text{pre-train}}$ | 32.00 | 33.58 | 41.32 |
| | Acc.$_{\text{fine-tune}}$ | 90.20 | 88.73 | 88.73 |
| STSB | Largest singular value | 0.93 | 1.40 | 1.08 |
| | Frobenius norm | 0.94 | 1.40 | 1.12 |
| | Eval. loss$_{\text{pre-train}}$ | 3.18 | 2.49 | 2.53 |
| | Acc.$_{\text{pre-train}}$ | 43.72 | 51.15 | 48.79 |
| | Acc.$_{\text{fine-tune}}$ | 91.16 | 91.02 | 91.03 |

As reported in Table 12, as $r$ changes, the largest singular value also varies, which, in turn affects the performance on the pre-trained task. The performance on the pre-trained task, however, does not degrade but rather shows an improvement. This indicates that the proposed model retains its ability to effectively mitigate catastrophic forgetting even as the rank increases.

## K ADDITIONAL EXPERIMENTS ON NATURAL LANGUAGE UNDERSTANDING

We conduct the experiments on the GLUE task with various LoRA-based methods applied to the DeBERTaV3$_{\text{base}}$, following the experimental environments in (Benedek & Wolf, 2024). The results are reported in Table 13.

Table 13: Performance comparison of various methods with DeBERTaV3$_{\text{base}}$ on GLUE tasks with 3 different random seeds. The results for the baselines are copied from (Benedek & Wolf, 2024).

| Method | CoLA | RTE | MRPC | STS-B |
|---|---|---|---|---|
| LoRA | 69.82 | 85.20 | 89.95 | 88.50 |
| AdaLoRA | 71.45 | 88.09 | 90.69 | 89.46 |
| PRILoRA | 72.79 | 89.05 | 92.49 | 90.01 |
| HiLoRA | **72.84** | **89.89** | **92.57** | **92.00** |

