# OpenReview forum: "HiLoRA: High-frequency-augmented Low-Rank Adaptation"
_ICLR.cc/2025/Conference — Submitted to ICLR 2025_

### Official Review · Reviewer_k5bg · 2024-10-19

**Soundness:** 3
**Presentation:** 3
**Contribution:** 3
**Rating:** 6
**Confidence:** 4

**Summary:**

The paper presents a method to mitigate the catastrophic forgetting of LoRA, by restricting the adaptation matrix to have singular values clamped to an upper bound.

**Strengths:**

* The authors tackle an important and meaningful topic that would be of interest to the community. PEFT is gaining more and more attention as the size of language models continues to grow.
* The results demonstrate that HiLoRA forgets less than LoRA, and some of its variants.

**Weaknesses:**

* Line 190, authors should explain more why depth of 2 or greater results in a separation of values.
* Equation (6), line 244, I believe the shape of U should be d_1xr, Sigma should be rxr and V should be rxd2, unlike what is written.
* Tables 1 and 2 better include the model name in the table description. The two tables use a different model.
* Table 1 shows that AdaLoRA has a lower average result than LoRA (On Roberta). However, in the original AdaLoRA paper that compared the two models on Deberta, AdaLora got better results. On other papers that compared the two on Roberta, AdaLora got better results. Are the results taken from other papers or calculated by the authors?
* Line 354, word Indicating should be with a small letter.
* Figure 4: In tables 1 and 2, authors compared 5 variants, and here only 4. AdaLoRA was omitted. The readers are interested to know the ‘forgetting’ property of AdaLoRA.

**Questions:**

* Equation (4), I can’t find the definition of i. Maybe it should appear as the index of the singular value on the left side, instead of n.
* Figure 4: The title says MRPC, however, in line 464 it says STS-B.

---

> ### Author Response · Authors · 2024-11-20
>
> We sincerely thank Reviewer K5bg for the review and feedback. Below, we would like to address each of your questions and concerns:
>
> **W1. Explanation of why a depth of 2 or greater results in a separation of singular values.**
>
>
> Proposition 3.1 in our paper cites Theorem 1 from the previous work [1], which states that gradient descent in deep networks implicitly drives the solutions towards low-rank.
>
> As revealed in [1], the pre-conditioning matrix $P$ serves as a preconditioning matrix to accelerate optimization by emphasizing an important directions (specific eigenvectors).
>
> For the dynamic optimizers like Adam, the dynamic preconditioning matrix $P_{W,G}$ is used. The eigenvalue of $P_{W,G}$ is derived as $(1 + \eta^2)^{-1/2}_{n,n'}$ and dynamically adjusted based on the magnitude of the gradient and the weight scale.
>
> In the directions with the large singular values, the gradient magnitude is relatively large, resulting in a smaller $\eta^2$, which enhances the contribution of those directions. Conversely, in the directions of the small singular values, $\eta^2$ becomes larger, suppressing learning towards those directions. As depth increases, the weighted combination of the preconditioning matrices across layers accumulates, further emphasizing the directions of the large singular values and the gap among singular values becomes more distinct.
>
> We have revised the paper with updated explanation.
>
> ---
>
> **W2. Typo.**
>
> Thanks for pointing that out. We have corrected the typo in the revised paper.
>
> ---
>
> **W3. Include model names in Tables 1 and 2.**
>
> Thank you for your thoughtful suggestion. While the backbone models used in the experiments are mentioned in the main text, we have also included them in Tables 1 and 2 to enhance clarity and understanding.
>
> ---
>
> **W4. Are the results taken from other papers or calculated by the authors?**
>
> For the GLUE task, we experimented with the baselines using four random seeds, following the recommended search range and including the best hyperparameters suggested in the original papers of each baseline. For the QA tasks, the results of the baselines were taken from AdaLoRA.
>
>
> ---
>
> **W5. Typo.**
>
> Thanks for pointing that out. We have corrected the typo in the revised paper.
>
> ---
>
> **W6. Catastrophic forgetting for AdaLoRA [2].**
>
> Following your kind suggestion, we measured the catastrophic forgetting phenomenon in AdaLoRA using MRPC and STS-B datasets for the GLUE task. Please refer the answer in **General Response 2**.
>
> ---
>
> **Q1 & Q2. Typos.**
>
> Thank you for bringing them to our attention. We have fixed the typo and incorporated the correction into the revised version of the paper.
>
> ---
>
> > [1] Zhao, Dan. "Combining explicit and implicit regularization for efficient learning in deep networks." *Advances in Neural Information Processing Systems* 35 (2022): 3024-3038.
> >
> > [2] Zhang, Qingru, et al. "AdaLoRA: Adaptive budget allocation for parameter-efficient fine-tuning." arXiv preprint arXiv:2303.10512 (2023).

---

> > ### Comment · Reviewer_k5bg · 2024-11-21
> > **Table 1 with DeBERTaV3-base**
> >
> > Thank you for the corrections.
> >
> > Many papers, like PRILoRA, ADALoRA and others, do the 'Table 1' comparison on DeBERTaV3-base. You did it on RoBERTabase.
> > Can you run HiLoRA for GLUE (8 tasks) on DeBERTaV3-base, and copy the results from the corresponding papers of ADALoRA, PRILoRA, LoRA? This would help the readers compare it with other papers. Running it should be simple, as you only need to change the backbone model.

---

> ### Author Response · Authors · 2024-11-27
>
> Thank you for reading our response.
>
> According to your reasonable suggestion, we conducted experiments on the GLUE task based on the DeBERTaV3$_{base}$ model to verify the performance of the proposed model. In accordance with your request, we copied the performance results of the baselines from the original paper of PRILoRA. For HiLoRA, we conducted experiments on 3 random seeds according to PRILoRA. Due to the time constraints, we first reported the results of 4 out of 8 datasets and revised the paper. We will report the results of the datasets that require long time for fine-tuning during the remaining period.
>
>
> | Method |  CoLA |  RTE | MRPC | STS-B |
> | --- | --- | --- |  --- |  --- |
> | LoRA | 69.82 |85.20| 89.95| 91.60 |88.50|
> | AdaLoRA | 71.45 | 88.09| 90.69| 91.84| 89.46|
> | PRILoRA  |  72.79| 89.05| 92.49 |91.92| 90.01|
> | HiLoRA  |  **72.84** | **89.89** | **92.57** | **92.00** |

---

> > ### Comment · Reviewer_k5bg · 2024-11-27
> >
> > Thank you. I believe it provides an interesting addition of information for the readers.
> >
> >
> >
> >
> >
> >
> > .

---

### Official Review · Reviewer_mcnZ · 2024-10-29

**Soundness:** 2
**Presentation:** 2
**Contribution:** 2
**Rating:** 5
**Confidence:** 4

**Summary:**

This paper addresses an intriguing issue—mitigating forgetting during fine-tuning of neural networks. It introduces a method focused on the fine-tuning of high-frequency components of pre-trained weights, claiming measurable improvements.

**Strengths:**

1.	This paper proposes a new sight for fine-tuning and tries to solve the forget problem in fine-tuning.
2.	It does reduce the accuracy loss on the pre-trained task.

**Weaknesses:**

1.The architecture diagrams in Figure 2 are unclear. To improve clarity, the implementation details of HiLoRA should be included in the main paper instead of relegated to the appendix.

2.The HiLoRA design lacks novelty.

3.The paper mentions, "U and V can be initialized with random r singular vectors of W0, or with U initialized to zero and V with a random Gaussian initialization." What initialization strategy was used in your experiments?

4.A deeper analysis of the relationship between the pre-trained task and the fine-tuning task in relation to the forgetting method would be beneficial. If the pre-trained and fine-tuning tasks are similar, does the forgetting problem still occur?

5.A scaling-up experiment, such as fine-tuning LLaMA-2-7B with Meta-Math and evaluating it on GSM8K, as in PiSSA, would be insightful.

6.Does this method remain effective when the rank increases?

7.Equations (6) and (7) should be introduced within Algorithm 1.

**Questions:**

see weaknesses

---

> ### Author Response · Authors · 2024-11-20
>
> We sincerely thank Reviewer mcnZ for the review and feedback. Below, we would like to address each of your questions and concerns:
>
> **W1. Figure 2 is unclear. The implementation details should be included in the main paper.**
>
> As you suggest, we include the implementation details in the main paper. Please refer the revised paper.
>
> Additionally, we would like to emphasize that Figure 2 shows the difference between the existing LoRA and our way of interpreting the adapter. In the existing LoRA, the adapter is interpreted as an adapter that is residual to $W_0$. However, we interpret the adapter as a component augmented with $W_0$, and show in Equation 13 that it is mathematically equivalent to the residual method.
>
> ---
>
> **W2. Lacks of novelty.**
>
> To highlight the novelty of HiLoRA, we present the novelty comparison table below.
>
> |  | How to interpret $\Delta W$ | How to regularize frequencies | Mitigate catastrophic forgetting |
> | --- | --- | --- |--- |
> | LoRA | Low-rank residual adapter | $\times$ | $\times$ |
> | PiSSA| Low-rank residual adapter | Initialize with the $r$ lowest-frequency components of $W_0$ and no restrictinons during fine-tuning | $\times$ |
> | MiLoRA | Low-rank residual adapter | Initialize with the $r$ highest-frequency components of $W_0$ and no restrictinons during fine-tuning | $\bigtriangleup$ |
> | HiLoRA | High-frequency augmented adapter | Restrict the frequencies of the newly added components to a certain threshold | $\bigcirc$ |
>
> LoRA and PiSSA primarily focus on improving the performance of fine-tuning tasks. In particular, PiSSA directly modifying the principal components, inducing significant changes to the pre-trained information, which leads to catastrophic forgetting.
>
> MiLoRA considers the high-frequency components as noise and directly modifies them. However, as revealed in Figure 1 in the main paper, the high-frequency components are not merely noise. Therefore, directly modifying the high-frequency components results in the loss of valuable information in MiLoRA. Moreover, since the frequency scale of the adapter of MiLoRA is not regularized during the fine-tuning process, the adapter’s information becomes dominant, causing catastrophic forgetting. Please refer to Section 5 for more details.
>
> In contrast, HiLoRA reinterprets the adapter as new high-frequency components augmented to $W_0$ and restricts their frequency range. Clipping the learnable diagonal-matrix $\Sigma$, which constains singular values, provides a simple yet highly effective method to adjust the frequency scale without additional losses or computational overheads. This approach confines the fine-tuning process to the high-frequency range while preserving the pre-trained information, which not only effectively mitigates catastrophic forgetting during fine-tuning but also achieving strong performance in downstream tasks.
>
>
> ---
>
> **W3. Initialization Method.**
>
> Thanks for your valuable feedback. We experimented with both initialization methods and found that there is an initialization method that is suitable for each data. Regardless of which initialization method is used, it eventually works as a singular vector of new components suitable for the down-stream task through orthogonal regularization. We have revised Tables 5 and 7 in Appendix to specify the initialization method used for each experiment.
>
> ---
>
> **W4. Does the forgetting problem still occur if the pre-trained and fine-tuning tasks are similar?**
>
> Thank you for your insightful comment.
>
> Generally, fine-tuning aims to inject the new knowledge based on the pre-trained information, which often leads to the catastrophic forgetting as a common phenomenon. If the pre-training and fine-tuning datasets are highly similar, the degree of catastrophic forgetting may not be significant. However, if there are differences in detailed information or class distributions between the datasets, catastrophic forgetting can still occur.
>
> In fact, Table 4 in the main paper reports the results of catastrophic forgetting for the proposed model on the GLUE task.  Within the GLUE task, MRPC dataset shows significant catastrophic forgetting in LoRA, while the SST-2 dataset exhibits relatively less forgetting. As you pointed out, the degree of catastrophic forgetting can vary depending on the dataset distribution. Nonetheless, to achieve the general capability  across diverse scenarios, it is crucial to design a model that can consistently mitigate catastrophic forgetting.

---

> > ### Author Response · Authors · 2024-11-20
> >
> > **W5. Scaling-up Experiments.**
> >
> > We conduct the large-scale experiment experiments for the commonsense reasoning task on LLaMA-7B. Please refer the answer in **General Response 3**.
> >
> > ---
> >
> > **W6. Effect of rank on catastrophic forgetting.**
> >
> > To verify whether HiLoRA maintains its performance and continues to mitigate forgetting as the rank increases, we conduct sensitivity study on the rank $r$ on MRPC and STS-B dataset. Specifically, we measured the largest singular value and Frobenius norm of the difference between the pre-trained model and the fine-tuned model. Also, we evaluated the accuracy and the evaluation loss on the pre-trained task, and accuracy on the fine-tuned task.
> >
> > - MRPC dataset
> >
> > |  | 8 | 16 | 64 |
> > |  --- | --- | --- | --- |
> > |  Largest singular value | 0.9283 | 0.6905 | 0.3583 |
> > |  Frobenius norm | 0.9358 | 0.6981 | 0.3626 |
> > |  Evaluation loss on pre-trained task | 4.3497 | 4.1798 | 3.1993 |
> > |  Accuray on pre-trained task | 32.00 | 33.58 | 41.32 |
> > |  Accuracy on fine-tuned task | 90.2 | 88.73 | 88.73 |
> >
> > - STS-B dataset
> >
> > |      | 8 | 16 | 64 |
> > | --- | --- | --- | --- |
> > |  Largest singular value | 0.9312 | 1.4038 | 1.0831 |
> > |  Frobenius norm | 0.9351 | 1.3991 | 1.1234
> > |  Evaluation loss on pre-trainedtask | 3.1785 | 2.4853 | 2.5323 ||
> > |  Accuray on pre-trained task | 43.72 | 51.15 | 48.79 |
> > |  Accuracy on fine-tuned task | 91.16 | 91.02 | 91.03 |
> >
> > According to the table, as $r$ changes, the largest singular value also varies, which, in turn affects the performance on the pre-trained task. The performance on the pre-trained task, however, does not degrade but rather shows an improvement. This indicates that the proposed model retains its ability to effectively mitigate catastrophic forgetting even as the rank increases.
> >
> > We have included the results in the revised paper.
> >
> > **W7. Equations (6) and (7) should be included within Algorithm 1.**
> >
> > Following your kind suggestion, we have incorporated Equations 6 and 7 into Algorithm 1. Please refer the revised paper.
> >
> > ---
> >
> > > [1] Hu, Zhiqiang, et al. "Llm-adapters: An adapter family for parameter-efficient fine-tuning of large language models." *arXiv preprint arXiv:2304.01933* (2023).

---

> > > ### Author Response · Authors · 2024-12-01
> > > **Gentle Reminder**
> > >
> > > Dear Reviewer mcnZ,
> > >
> > > We have made much effort to address all the questions and comments you provided in the review. If there are any follow-up questions or additional concerns, please do not hesitate to let us know. We would be happy to provide further clarification.
> > >
> > > Thank you for your time and consideration.
> > >
> > > Best,
> > >
> > > Authors

---

### Official Review · Reviewer_vQ3k · 2024-11-03

**Soundness:** 1
**Presentation:** 2
**Contribution:** 1
**Rating:** 3
**Confidence:** 4

**Summary:**

In this paper, the author proposes a new method to address the problem of catastrophic forgetting in LoRA fine-tuning. They suggest learning the adapter $\Delta W$ with small eigenvalues and validate their method across various downstream tasks.

**Strengths:**

1.	This paper describes the problem and method in detail.
2.	The authors validate the effectiveness on various downstream datasets.

**Weaknesses:**

1.	The definition of "frequency" in the paper is confusing and meaningless; low frequency and high frequency merely refer to the value of the eigenvalues. If the largest eigenvalue corresponds to pre-trained knowledge, why does an increase in the largest singular value during fine-tuning, as shown in Figure 1(a), lead to a decline in performance on the pre-training task? From Figure 1(a), it appears that they are either negatively correlated or unrelated.
2.	The novelty is limited. It is very similar to, including the implementation in Figure 7 and Equation (8). The only difference is that the $\Sigma$ matrix is learnable and clipped according to Equation (7). Additionally, in Table 2, HiLoRA performs worse than AdaLoRA, and the authors do not address the issue of forgetting in AdaLoRA.
3.	The motivation for investigating catastrophic forgetting in LoRA requires further explanation. When employing LoRA, my primary concern is its performance on specific downstream tasks. If I need to tackle multiple different tasks, I would prefer to use a general LLM or load various LoRA adapters through MultiLoRA.
4.	More comparisons are needed. The authors should include the results of PiSSA and MiLoRA in Table 2, and it needs to compare with methods such as DoRA, rsLoRA, and LoRA+. Furthermore, experiments should be conducted on larger LLM models, such as LLaMA.

**Questions:**

See Weaknesses.

---

> ### Author Response · Authors · 2024-11-20
>
> We sincerely thank Reviewer vQ3k for the review and feedback. Below, we would like to address each of your questions and concerns:
>
> **W1. The connection between frequency, largest singular value, and catastrophic forgetting.**
>
> We apologize for the confusion in our explanation regarding the relationship among the frequency, singular value, and catastrophic forgetting.
>
> In the field of signal processing, "frequency" is closely related to the information contained in a weight $W$. Typically, the low-frequency components of $W$ i) correspond to large singular values of $W$ [1, 2] and ii) are used to capture the global information (or decide the global pattern of $W$). Conversely, high-frequency components are i) associated with small singular values and ii) used to capture remaining fine-grained details [3,4,5]. When the adapter learns mainly for low-frequency components, the model tends to include a significant amount of the global information from the fine-tuning dataset/task. However, if the global information from the fine-tuning dataset/task becomes overly dominant, the information learned during pre-training will be overwhelmed, leading to a degradation in the performance of the pre-training task.
>
> We regard the information from the fine-tuning task is rather fine-grained details in comparison with the pre-trained knowledge, considering their dataset size difference. Therefore, we naturally let our proposed method focus on learning the new task by using only the high-frequency components. As described in Section 5 (Analyses on HiLoRA), our model demonstrates the ability to maintain the pre-training task performance by learning the adapters only in the high-frequency domain, i.e., the small singular value range.
>
> In addition, in the transfer learning where preventing catastrophic forgetting is crucial, a common regularization technique is Elastic Weight Consolidation (EWC) [6,7]. EWC introduces an additional regularization term to the loss function to penalize deviations from the pre-trained weight. In other words, EWC only perfers adjusting high-frequency components since adjusting low-frequency components changes the global pattern of the pre-trained weight, i.e., a large deviation. Our method directly adapts in the singular value domain, i.e, in the frequency domain, which is more effective in achieving the goal.

---

> ### Author Response · Authors · 2024-11-20
>
> **W2. i) Limited Novelty. ii) Comparison with AdaLoRA.**
>
> i) To highlight the novelty of HiLoRA, we present the novelty comparison table below.
>
> |  | How to interpret $\Delta W$ | How to regularize frequencies | Mitigate catastrophic forgetting |
> | --- | --- | --- |--- |
> | LoRA | Low-rank residual adapter | $\times$ | $\times$ |
> | PiSSA| Low-rank residual adapter | Initialize with the $r$ lowest-frequency components of $W_0$ and no restrictinons during fine-tuning | $\times$ |
> | MiLoRA | Low-rank residual adapter | Initialize with the $r$ highest-frequency components of $W_0$ and no restrictinons during fine-tuning | $\bigtriangleup$ |
> | HiLoRA | High-frequency augmented adapter | Restrict the frequencies of the newly added components to a certain threshold | $\bigcirc$ |
>
> LoRA and PiSSA primarily focus on improving the performance of fine-tuning tasks. In particular, PiSSA directly modifying the principal components, inducing significant changes to the pre-trained information, which leads to catastrophic forgetting.
>
> MiLoRA considers the high-frequency components as noise and directly modifies them. However, as revealed in Figure 1 in the main paper, the high-frequency components are not merely noise. Therefore, directly modifying the high-frequency components results in the loss of valuable information in MiLoRA. Moreover, since the frequency scale of the adapter of MiLoRA is not regularized during the fine-tuning process, the adapter’s information becomes dominant, causing catastrophic forgetting. Please refer to Section 5 for more details.
>
> In contrast, HiLoRA reinterprets the adapter as new high-frequency components augmented to $W_0$ and restricts their frequency range. Clipping the learnable diagonal-matrix $\Sigma$, which constains singular values, provides a simple yet highly effective method to adjust the frequency scale without additional computational overheads. This approach confines the fine-tuning process to the high-frequency range while preserving the pre-trained information, which not only effectively mitigates catastrophic forgetting during fine-tuning but also achieving strong performance in downstream tasks.
>
> ii) Comparison with AdaLoRA
>
> HiLoRA shows improved performance compared to AdaLoRA on the GLUE task and achieved more or less the same performance on average for the QA task. However, as can be seen from the answer in **General Response 2**, AdaLoRA suffers from catastrophic forgetting as fine-tuning progresses, while HiLoRA effectively mitigates this phenomenon. This suggests that HiLoRA not only delivers promising performance but also excels in preserving existing knowledge, which demonstrates the enhanced expressiveness and generalization capabilities.
>
> ---
>
> **W3. Why catastrophic forgetting is important in LoRA?**
>
> LLMs are widely used because they are equipped with rich expressiveness and generalization capabilities, achieved by training large-scale models on vast language datasets. In contrast, the datasets used for fine-tuning are relatively much small in size. Catastrophic forgetting after fine-tuning refers to the phenomenon where the model forgets the knowledge acquired during the pre-training phase, resulting in the generalization capability degradation. In other words, the model becomes overfitted to the target fine-tuning dataset/task, losing its generalization capabilities. In real-world environments, however, we cannot foresee future user queries so mainingtaining the generalization capability and achievning a high fune-tuning task accuracy are equally important. Consequently, recent studies, such as those in [8, 9], have focused on addressing catastrophic forgetting in the context of fine-tuning LLMs, particularly with methods like LoRA.
>
> Therefore, mitigating catastrophic forgetting is as important as the fine-tuning task accuracy from the perspective of model reliability and scalability. In other words, addressing catastrophic forgetting in LoRA is crucial as it ensures that the strengths of pre-trained models are preserved while efficiently adapting to new tasks and delivering consistent performance.
>
> LoRA is designed for modularity, allowing incremental updates for new tasks without retraining the full model. Catastrophic forgetting makes this approach less effective, as the model may fail to retain prior knowledge while learning new tasks, hindering scalability in multi-task or continual learning scenarios.
>
> Furthermore, we demonstrated improved performance and effective mitigation of catastrophic forgetting through experiments in Section 4 (Experiments) and Section 5 (Analyses on HiLoRA).
>
> ---
>
> **W4. Experiments on various baselines and large-scale models.**
>
> Following your kind suggestion, we have added baselines for the models you mentioned on the GLUE task. Please refer the answer in **General Response 1**.
>
> Furthermore, we conduct the large-scale experiments for the commonsense reasoning task on LLaMA-7B. Please refer the answer in **General Response 3**.

---

> ### Author Response · Authors · 2024-11-20
>
> References
>
> > [1] Chen, Yang et al., "Graph fourier transform based on singular value decomposition of the directed laplacian." Sampling Theory, Signal Processing, and Data Analysis, 2023.
> >
> > [2] Maskey, et al., "A fractional graph laplacian approach to oversmoothing." Advances in NeurIPS, 2024.
> >
> > [3] Cooley et al., "The fast Fourier transform and its applications." IEEE Transactions on Education, 1969.
> >
> > [4] Deng et al., "An adaptive Gaussian filter for noise reduction and edge detection." IEEE conference record nuclear science symposium and medical imaging conference, 1993.
> >
> > [5] Pan et al., "Fast vision transformers with hilo attention." Advances in NeurIPS, 2022.
> >
> > [6] Kirkpatrick et al., "Overcoming catastrophic forgetting in neural networks." Proceedings of the national academy of sciences, 2017.
> >
> > [7] Kemker et al., "Measuring catastrophic forgetting in neural networks." Proceedings of the AAAI conference on artificial intelligence, 2018.
> >
> > [8] Fu et al.,"LoFT: LoRA-Based Efficient and Robust Fine-Tuning Framework for Adversarial Training." International Joint Conference on Neural Networks (IJCNN). IEEE, 2024.
> >
> > [9] Chen et al., "Bayesian Parameter-Efficient Fine-Tuning for Overcoming Catastrophic Forgetting." preprint, 2024.

---

> ### Author Response · Authors · 2024-11-23
> **Gentle Reminder**
>
> Dear Reviewer vQ3k,
>
> We have made every effort to address all the questions and comments you provided in the review. If there are any follow-up questions or additional concerns, please do not hesitate to let us know. We would be happy to provide further clarification.
>
> Thank you for your time and consideration.

---

> ### Comment · Reviewer_vQ3k · 2024-11-23
>
> I apologize for omitting the term "AdaLoRA" in W2. Allow me to restate my question for clarity.
>
> HiLoRA appears to be very similar to AdaLoRA, particularly in the implementation details presented in Figure 7 and Equation (8). The main difference seems to be that the $\Sigma$ matrix in HiLoRA is learnable and clipped according to Equation (7).

---

> ### Author Response · Authors · 2024-11-25
>
> |  | AdaLoRA | HiLoRA |
> | -------- | -------- | -------- |
> | Motivation | Improve parameter efficiency| Mitigate catastrophic forgetting \& Improve parameter efficiency |
> | Rank of adapter $r$   | Dynamically adjusted with the importance score based on the magnitude of singular values or the sensitivity of the loss w.r.t. the parameter  | Fixed as a small pre-defined value |
> | How to regularize frequencies | None | Restrict the frequencies of the newly added components to a certain threshold |
>
> In recent years, many LoRA-based studies have proposed various distinctions based on the parameterized SVD. **Figure 7 and Equation (8) represent the common framework of all the parameterized SVD-based LoRA.**
>
> For instance, LoRA$^2$ [1] uses the twice-nested parameterized SVD to iteratively project the token representations onto mutually orthogonal planes. SORSA [2], similar to PiSSA, achieves better convergence by initializing the adapter with $r$ principal components from the pre-trained weight and fine-tuning it with orthogonal regularization on the singular vectors. Mo-SARA [3] initializes the singular vectors with those of $r$ principal components, freezes the singular vectors, and fine-tunes the randomly initialized multiple singular values. These singular values are then gated through a learnable vector. Therefore, many studies, which were designed based on the parameterized SVD, i.e., Equation (8), differ in their specific design strategies.
>
> In this context, both AdaLoRA and HiLoRA use the parameterized SVD, but they differ in their motivations, design, and implementation details.
>
> AdaLoRA focuses on the parameter efficiency, dynamically adjusting the rank of each layer within a constrained parameter budget to learn the optimal rank. Each singular value is assigned an importance score based on the magnitude of singular values or the sensitivity of the loss w.r.t. the parameter, and less important singular values are dynamically pruned to optimize the parameter usage. Consequently, the final rank for each layer may differ, but the lack of consideration for frequency components can still lead to catastrophic forgetting as shown in our **General Response #2**.
>
> In terms of implementation, **AdaLoRA computes an importance score for each singular value, and keeps singular values whose scores exceed a threshold, and sets the others to zero**.
>
> On the other hand, HiLoRA emphasizes enhancing expressiveness and mitigating catastrophic forgetting at the same time. HiLoRA restricts the frequency domain of the adapter to high frequency within a fixed rank.
>
> Its implementation of clipping ensures that **the singular values are restricted below a predefined upper limit using clipping. It ensures that the singular values are learned in the high-frequency domain. Unlike AdaLoRA, HiLoRA does not calculate importance scores or prune the singular values to zero.** These implementation differences may appear insignificant, but they have a profound impact on preventing the catastrophic forgetting.
>
> In summary, although many methods use the parameterized SVD, their motivations and implementation approaches are fundamentally different. Future research could explore combining both methods to dynamically adjust ranks within high-frequency domains, offering a novel approach to fine-tuning.
>
>
> > [1] Zhang et al., "LoRA $^ 2$: Multi-Scale Low-Rank Approximations for Fine-Tuning Large Language Models." arXiv preprint, 2024.
> >
> > [2] Cao et al., "Sorsa: Singular values and orthonormal regularized singular vectors adaptation of large language models." arXiv preprint, 2024.
> >
> > [3] Gu et al., "Sara: Singular-value based adaptive low-rank adaption." arXiv preprint, 2024.

---

> > ### Author Response · Authors · 2024-12-01
> > **Gentle Reminder**
> >
> > Dear Reviewer vQ3k,
> >
> > We hope that our response has addressed your concerns adequately. If there are any unresolved issues or additional questions, please feel free to let us know. We would be happy to engage in discussion and address them. Thank you once again for your efforts.
> >
> > Best,
> >
> > Authors

---

> > > ### Comment · Reviewer_vQ3k · 2024-12-03
> > >
> > > Thank you for your reply and the additional experiments you included. After reviewing the feedback from the other reviewers, I have decided to keep my original rating.

---

### Official Review · Reviewer_36is · 2024-11-04

**Soundness:** 3
**Presentation:** 3
**Contribution:** 3
**Rating:** 6
**Confidence:** 4

**Summary:**

In this paper, the authors proposed a new parameter-efficient fine-tuning (PEFT) approach with learnable high-frequency components while  constraining singular values to appropriate levels. Multiple experiments and ablation studies were conducted to show the effectiveness of the proposed method.

**Strengths:**

1. The motivation and formulation of the proposed HiLoRA makes sense and is technically sound to me.
2. The authors conducted extensive experiments on multiple datasets and showed improved performance over several baseline methods.
3. Ablation studies and sensitivity analysis were also conducted to show the effectiveness of the proposed approach.
4. Writing is good and easy to follow.

**Weaknesses:**

1. The performance improvement seems very small as shown in table 1 and 2 and sometimes even worse than baseline methods.
2. Are there any principles or rule or thumb for setting those hyper-parameters for different tasks/models?

**Questions:**

Overall, I think the HiLoRA proposed in this paper is novel and beneficial to the community. Please refer to the weakness section to prepare rebuttals on the performance and hyper-parameter setting.

---

> ### Author Response · Authors · 2024-11-20
>
> We sincerely thank Reviewer 36is for the review and positive feedback. Below, we would like to address each of your questions and concerns:
>
> **W1. Performance improvement is marginal.**
>
> While our model has consistently surpassed various baselines on average, we conduct additional experiments with state-of-the-art baselines for the GLUE task to further validate the effectiveness of our proposed method and report the results in the below table. Due to time constraints, we have included these baselines only for the following six datasets, and we plan to continue updating the tables for other two remaining datasets.
>
> | Method | CoLA | QNLI | RTE | MRPC | STS-B |
> | --- | --- | --- | --- | --- | --- |
> | DoRA [1] | 64.29 | 92.90 | 81.68 | 89.52 | 90.95 |
> | LoRA+ [2] | 63.63 | 92.72 | 81.23 | 89.03 | 90.87 |
> | rsLoRA [3] | 64.17 | 92.59 | 79.78 | 89.40 | 90.84 |
> | HiLoRA | **64.66** | **93.12** | **82.85** | **90.20** | **91.16** |
>
> Across these diverse baselines, our model still demonstrated improved performance on average. One may consider that our improvements are marginal but in fact, our method provides larger improvements than others.
>
> Furthermore, our core contribution is that, by reinterpreting the adapter from a signal processing perspective, we showed that a simple mechanism for controlling singular values allows our model to maintain the expressive power while effectively mitigating catastrophic forgetting compared to the baselines.
>
> ---
>
> **W2. Principle for searching hyperparameters.**
>
> We conducted a grid search primarily around the values commonly used in LoRA-based methods. The range of hyperparameters we explored can be found in Appendix E.1.2 and E.2.2.
>
>
> ---
>
> In order to validate the **effectiveness and generalizability** of our model, we conducted the following additional experiments, the results of which can be found in the General Response and the revised paper:
>
> 1. Catastrophic forgetting in AdaLoRA
> 2. Validation of the scalability of the proposed method through large-scale experiments.
>
>
>
> >[1] Liu, Shih-yang, et al. "DoRA: Weight-Decomposed Low-Rank Adaptation." *Forty-first International Conference on Machine Learning*.
> >
> >[2] Hayou, Soufiane, Nikhil Ghosh, and Bin Yu. "Lora+: Efficient low rank adaptation of large models." *arXiv preprint arXiv:2402.12354* (2024).
> >
> >[3] Kalajdzievski, Damjan. "A rank stabilization scaling factor for fine-tuning with lora." *arXiv preprint arXiv:2312.03732* (2023).

---

> > ### Comment · Reviewer_36is · 2024-11-21
> >
> > Thanks for the response. Most of my main concerns are addressed and thus I'll keep my original positive rating.

---

> ### Author Response · Authors · 2024-11-27
>
> Dear Reviewer 36is,
>
> We thank the reviewer for recognizing the novelty and contributions of HiLoRA to the community. We are glad that we could address your concerns and appreciate your continued positive rating. However, we believe there may still be additional considerations that could further improve the evaluation. We would be grateful for the opportunity to address these considerations.

---

### Author Response · Authors · 2024-11-20
**General Response to Reviewers**

We thank all reviewers for their valuable time and insightful comments. We leave key general remarks here and will respond to each review below.

**1. Comparison with other state-of-the-art baselines. (Reviewer 36is, vQ3k)**

we have added state-of-the-art baselines on the GLUE task with 4 random seeds. Due to time constraints, we have included these baselines only for the following six datasets, and we plan to continue updating the tables for other two remaining datasets.

| Method | SST2 |CoLA | QNLI | RTE | MRPC | STS-B |
| --- | --- | --- | --- | --- | --- | --- |
| DoRA [1] | 92.32 |64.29 | 92.90 | 81.68 | 89.52 | 90.95 |
| LoRA+ [2] | 93.98 |63.63 | 92.72 | 81.23 | 89.03 | 90.87 |
| rsLoRA [3] | 95.04 |64.17 | 92.59 | 79.78 | 89.40 | 90.84 |
| HiLoRA | **95.10** | **64.66** | **93.12** | **82.85** | **90.20** | **91.16** |

**2. Catastrophic forgetting in AdaLoRA. (Reviewer vQ3k, K5bg)**

We measured the catastrophic forgetting phenomenon in AdaLoRA using the MRPC and STS-B datasets in the GLUE task. Specifically, we measured the largest singular value and Frobenius norm of the difference between the pre-trained model and the fine-tuned model. Also, we evaluated the accuracy and the evaluation loss on the pre-trained task. For each dataset, the metrics were measured every 5 epochs during the fine-tuning of AdaLoRA, and the metrics at the point where AdaLoRA and HiLoRA achieved their best accuracy are also reported, respectively. The results are presented in the table below.

- MRPC dataset

| Epoch | 5  | 10 | 15 | 20 | 25 | 30 | Best (AdaLoRA) | Best (HiLoRA) |
| --- | --- | --- | --- | --- | --- | --- | --- | --- |
| Largest singular value | 2.0177 | 2.0069 | 2.0016 | 2.0009 | 2.0004 | 2.0001 | 2.0177 | 0.9358 |
| Frobenius norm | 2.0201 | 2.0050 | 2.0014 | 2.0011 | 2.0005 | 2.0001 | 2.0201 | 0.9284 |
| Accuray on pre-trained task  | 25.578 | 14.879 | 8.962 | 16.794 | 10.591 | 11.343 | 25.58 | 32.00 |
| Evaluation loss on pre-trained task | 5.231 | 6.696 | 7.617 | 6.178 | 7.103 | 7.007 | 5.2308 | 4.3496 |

- STS-B dataset

| Epoch | 5 | 10 | 15 | 20 | 25 |  Best (AdaLoRA) | Best (HiLoRA) |
|  --- | --- | --- | --- | --- | --- | --- | --- |
| Largest singular value | 2.0064 | 2.0034 | 2.0056 | 2.0021 | 2.0001 | 2.0001 | 0.9351 |
| Frobenius norm | 2.0087 | 2.0030 | 2.0065 | 2.0013 | 2.0001 | 2.0001 | 0.9312 |
| Accuray on pre-trained task  | 33.526 | 34.118 | 29.61 | 28.19 | 28.485 | 28.49 | 43.72 |
| Evaluation loss on pre-trained task | 3.984 | 3.951 | 4.426 | 4.566 | 4.525 | 4.5248 | 3.1785 |

According to the tables, AdaLoRA also experiences catastrophic forgetting as its fine-tuning progresses. The Frobenius norm increases from 0 to 2 in the early fine-tuning phase, with the performance on the pre-trained task decreases. Even at its peak performance during fine-tuning, the model still exhibits low performance on the pre-trained task.

This can be attributed to the lack of consideration for frequency components of adapters, leading to a tendency for learning the low-frequency components while forgetting the pre-trained information. In contrast, the proposed model regularizes the frequency components in the adapter, injecting the new knowledge into high-frequency components during fine-tuning. As a result, the proposed model retains pre-trained information more effectively.

**3. Validation of the scalability of the proposed method through large-scale experiments. (Reviewer vQ3k, K5bg)**

We conduct the experiments for the commonsense reasoning task on LLaMA-7B. Following [4], we amalgamate the training datasets from 8 sub-tasks to create the final training dataset, and conduct evaluations on the individual testing dataset for each task. The results are provided in below table.


| LLaMA-7B | BoolQ | PIQA  | SIQA  | HellaSwag | WinoGrande | ARC-e | ARC-c | QBQA  | Avg.  |
|----------|-------|-------|-------|-----------|------------|-------|-------|-------|-------|
| LoRA     | 68.9  | 80.7  | 77.4  | 78.1      | 78.8       | 77.8  | 61.3  | 74.8  | 74.7  |
| HiLoRA | 62.2 | 82.7 | 78.3 | 81.0 | 80.9 | 83.3 | 66.8 | 78.6 | **76.7** |

HiLoRA demonstrates improved performance over LoRA on average in large-scale models, highlighting its stability and effectiveness while maintaining strong results across diverse downstream tasks.


>[1] Liu, Shih-yang, et al. "DoRA: Weight-Decomposed Low-Rank Adaptation." Forty-first International Conference on Machine Learning.
>
>[2] Hayou, Soufiane, Nikhil Ghosh, and Bin Yu. "Lora+: Efficient low rank adaptation of large models." preprint, 2023.
>
>[3] Kalajdzievski, Damjan. "A rank stabilization scaling factor for fine-tuning with lora."  preprint, 2023.
>
>[4] Hu et al. "LLM-adapters: An adapter family for parameter-efficient fine-tuning of large language models." preprint, 2023.

---

### Comment · Area_Chair_UfhB · 2024-11-22
**Interactive Discussions**

Dear Reviewers,

Thank you for your efforts in reviewing this paper. We highly encourage you to participate in interactive discussions with the authors before November 26, fostering a more dynamic exchange of ideas rather than a one-sided rebuttal.

Please feel free to share your thoughts and engage with the authors at your earliest convenience.

Thank you for your collaboration.

Best regards,
ICLR 2025 Area Chair

---

### Meta-Review · Area_Chair_UfhB · 2024-12-19

**Metareview:**

This submission proposes a simple method to address catastrophic forgetting in LoRA fine-tuning. The method focuses on learning adapters with small eigenvalues, with its performance validated across various downstream tasks.

Strengths: Simple method, promising performance.

Weaknesses: Limited novelty, over-claimed forgetting mitigation.

After reviewing all the feedback (2 positive and 2 negative reviews) and the rebuttals, the Area Chair concludes that the submission does not meet the ICLR standard (borderline on the reject side) mainly due to its lack of novelty compared to existing LoRA fine-tuning methods.

**Additional Comments On Reviewer Discussion:**

While many concerns were addressed in the rebuttal, one important issue—the novelty of the proposed method compared to existing LoRA fine-tuning methods—remains unresolved.

---

### Decision · Program_Chairs · 2025-01-22

Reject